

# The riverine source of tropospheric CH$_4$ and N$_2$O from the Republic of Congo, Western Congo Basin

Robert C. Upstill-Goddard[1], Matthew E. Salter[2], Paul J. Mann[3], Jonathan Barnes[1], John Poulsen[4], Bienvenu Dinga[5], Gregory J. Fiske[6], and Robert M. Holmes[6]

[1]Oceans and Climate Research Group, School of Marine Science and Technology, Newcastle University, NE1 7RU, UK.
[2]Department of Environmental Science and Analytical Chemistry, Stockholm University, 11418 Stockholm, Sweden
[3]Department of Geography, Northumbria University, NE1 8ST, UK.
[4]Nicholas School of the Environment, Duke University, PO Box 90328, Durham, NC, 27708, USA
[5] Laboratoire de Physique de l'Atmosphère, Université de Marien Ngouabis, Brazzaville, B.P. 69, Republic of Congo.
[6]Woods Hole Research Center, 149 Woods Hole Road, Falmouth, MA 02540-1644 USA.

*Correspondence to:* Robert C. Upstill-Goddard (rob.goddard@ncl.ac.uk)

**Abstract.** We report concentrations of dissolved CH$_4$, N$_2$O, O$_2$, NO$_3^-$ and NH$_4^+$, and corresponding CH$_4$ and N$_2$O emissions for river sites in savanna, swamp forest and tropical forest, along the Congo main stem and in several of its

tributary systems of the Western Congo Basin, Republic of Congo, during November 2010 (41 samples; "wet season") and August 2011 (25 samples; "dry season"; CH$_4$ and N$_2$O only). Dissolved inorganic nitrogen (DIN: wet season; NH$_4^+$ + NO$_3^-$) was dominated by NO$_3^-$ (63 ± 19% of DIN), total DIN concentrations (1.5-45.3 µmol L$^{-1}$) being consistent with small agricultural, domestic and industrial sources. Dissolved O$_2$ (wet season) was mostly under-saturated in swamp forest (36 ± 29%) and tropical forest (77 ± 36%) rivers but predominantly super-saturated in

savannah rivers (100 ± 17%). Dissolved CH$_4$ and N$_2$O were within previously reported ranges for sub-Saharan African rivers. While CH$_4$ was always super-saturated (11.2 - 9553 nmol L$^{-1}$; 440-354400%), N$_2$O ranged from strong under-saturation to strong super-saturation (3.2-20.6 nmol L$^{-1}$; 47-205%). Evidently, rivers of the ROC are persistent local sources of tropospheric CH$_4$ but can be small sources or sinks for N$_2$O. Dry season concentration means and ranges of CH$_4$ and N$_2$O were indistinguishable for all three land types and seasonal differences in means and ranges

were not significant for N$_2$O for any land type or for CH$_4$ in savannah rivers; the latter is consistent with seasonal buffering of river discharge by an underlying sandy-sandstone aquifer. By contrast, swamp and forest river CH$_4$ was significantly higher in the wet season, possibly reflecting CH$_4$ derived from floating macrophytes during flooding and/or enhanced methanogenesis in adjacent flooded soils. Swamp rivers exhibited both low (47%) and high (205%) N$_2$O saturations but wet season values were overall significantly lower than in either tropical forest or savannah rivers,

which were always super-saturated (103-266%) and for which the overall means and ranges of N$_2$O were not significantly different. In swamp and forest rivers % O$_2$ co-varied negatively with log % CH$_4$ and positively with % N$_2$O. The strong positive N$_2$O - O$_2$ correlation in swamp rivers was coincident with strong N$_2$O and O$_2$ under-saturation, indicating N$_2$O consumption by sediment denitrification. In savannah rivers persistent N$_2$O super-saturation and a negative N$_2$O - O$_2$ correlation may indicate N$_2$O production mainly by nitrification, consistent with a

stronger correlation between N$_2$O and NH$_4^+$ than between N$_2$O and NO$_3^-$. Our range in CH$_4$ and N$_2$O emissions fluxes (33-48705 µmol CH$_4$ m$^{-2}$ d$^{-1}$; 1-67 µmol N$_2$O m$^{-2}$ d$^{-1}$), is wider than previously estimated for sub-Saharan African rivers but it includes uncertainties deriving from our use of "basin-wide" values for CH$_4$ and N$_2$O gas transfer velocities. Even so, because we did not account for any contribution from ebullition, which for CH$_4$ is likely to be at least 20%, our emissions estimates for CH$_4$ are probably conservative.

# 1 Introduction





Methane ($CH_4$) and nitrous oxide ($N_2O$) accounted for 17% and 6% respectively, of the total atmospheric radiative forcing by well-mixed greenhouse gases in 2011 (Myhre et al., 2013). $CH_4$ also impacts tropospheric oxidising capacity, $O_3$ and OH radical and is a source of stratospheric $O_3$ (Hartmann et al., 2013) while $N_2O$ is the largest cause of stratospheric $O_3$ loss, via NO production (Ravishankara et al., 2009). Since the onset of the industrial revolution, tropospheric $CH_4$ and $N_2O$ have substantially increased but their growth rates have varied. Following periods of

declining and zero growth from the mid 1980's, tropospheric $CH_4$ progressively increased from the late 2000's (Rigby et al., 2008; Dlugokencky et al., 2009). Growth ~ 4-5 ppbv $yr^{-1}$ since 2009 (Sussmann et al., 2012) has been linked to increasing natural tropical emissions (Bousquet et al., 2011). The mean $CH_4$ tropospheric dry mole fraction in 2011, $1803 \pm 2$ ppbv, was more than 150% above the pre-industrial value (Ciais et al., 2013). Increasing tropospheric $N_2O$ largely reflects its enhanced emission from soils, freshwaters and coastal waters via the accelerated mobilisation of

reactive nitrogen (Syakila and Kroeze, 2011). Consequently the IPCC now classifies river, estuary and coastal zone $N_2O$ sources as anthropogenic (Ciais et al., 2013). A small but significant seasonal to inter-annual variability in $N_2O$ growth rate may reflect climate-driven changes in soil $N_2O$ (Thompson et al., 2013). The current rate of $N_2O$ growth is $0.73 \pm 0.03$ ppb $yr^{-1}$ and its tropospheric dry mole fraction in 2011, $324 \pm 0.1$ ppbv, was ~20% above its pre-industrial value (Ciais et al., 2013).


The evidence base for freshwater ecosystems (streams, rivers, lakes, and reservoirs) as important sources of tropospheric $CH_4$ and $N_2O$ is small but increasing. The global freshwater $CH_4$ source could be ~$10^{13}$-$10^{14}$ g $yr^{-1}$ (Bastviken et al., 2011; Kirschke et al., 2013), the uncertainty reflecting data gaps, notably for major world river basins, and a sampling bias that has necessitated upscaling from exclusively temperate data (Bastviken et al., 2011).

Notwithstanding the uncertainty, this freshwater source estimate is ~30-47 % of natural $CH_4$ emissions and ~12-20% of total $CH_4$ emissions (Kirschke et al., 2013). Converting it to $CO_2$ equivalents based on warming potentials and atmospheric lifetimes gives $0.65 \times 10^{15}$ g C ($CO_2$ equivalent) $y^{-1}$ (Bastviken et al., 2011), a significant offset to the combined terrestrial and oceanic carbon sink ~ $3.5 \times 10^{15}$ g C $yr^{-1}$ (Le Quéré et al., 2015). A global estimate of river $N_2O$ emissions based on microbial production from agriculturally-derived nitrogen is ~ $6.8 \times 10^{11}$ g $yr^{-1}$, around 10%

of the total global anthropogenic $N_2O$ source, but because this involved upscaling emissions from entirely within the contiguous United States (Beaulieu et al., 2011), it too must be highly uncertain.

Tropical river systems in Africa include some of the world's largest, together contributing ~12% of both global freshwater discharge (Valentini et al., 2014) and river surface area (Raymond et al., 2013). Borges et al (2015b)

recently reported seasonal emissions ~$3-4 \times 10^{12}$ g $CH_4$ $yr^{-1}$ and ~$10^{10}$ g $N_2O$ $yr^{-1}$ for twelve large river systems in sub-Saharan Africa, including the three largest by catchment area (Congo, Niger, Zambezi). Notably, their $CH_4$ estimate is 5 times higher than was previously attributed to all tropical rivers (Bastviken et al., 2011) and both estimates are significant at the continental scale given that reported total African emissions are ~ $66 \pm 35 \times 10^{12}$ g $CH_4$ $yr^{-1}$ and $3.3 \pm 1.3 \times 10^{12}$ g $N_2O$ $yr^{-1}$ (Valentini et al., 2014).


The potential scale of $CH_4$ and $N_2O$ emissions from tropical freshwaters and their attendant uncertainties warrant further investigation. In this paper we present and discuss concentrations of dissolved $CH_4$, $N_2O$, $O_2$, $NO_3^-$ and $NH_4^+$, and corresponding $CH_4$ and $N_2O$ emissions for river sites in savanna, swamp forest and tropical forest, along the Congo main stem and in several of its tributary systems of the Western Congo Basin, Republic of Congo, during

November 2010 and August 2011.



## 2 Study site and sample locations

The ~4700 km long Congo River (Fig. 1) has an equatorial location that affords it a bimodal hydrological regime, with maximal flows in December and May and minimal flows in August and March (Coynel et al., 2005).   The Congo Basin (9°N - 14°S; 11° - 31°E) is the largest hydrological system in Central Africa, covering ~$3.8 \times 10^6$ km$^2$ (~ 12% of the total African land mass; Fig. 1) and incorporating the world's fourth largest wetland area ~$3.6 \times 10^5$ km$^2$ (Laporte et al., 1998). The Congo's annual freshwater discharge is the world's second largest at ~1300 km$^3$ (Borges et al., 2015b), 50% of all freshwater flow from Africa to the Atlantic Ocean.   Rivers and streams in the Congo Basin have a total open water surface area ~$2.7 \times 10^4$ km$^2$ (Raymond et al., 2013).   The climate is warm (mean annual temperature $24.8 \pm 0.8$ °C) and humid with an annual rainfall ~1800 mm (Laraque et al., 2001).

We sampled the Congo main stem, several of its tributary rivers and some of their sub-tributaries, at sites within the Republic of Congo (ROC: area $3.4 \times 10^5$ km²), in the western Congo Basin (Figure 1).   Individual catchment areas, freshwater discharge rates and rainfall are listed in Table 1.   Around 50% of the ROC land area is classified as tropical forest, with the remainder classified as either swamp or savannah in approximately equal proportion (Clark and Decalo, 2012).   Sampling sites were selected to represent each of these three land cover types (Fig. 1), which were georeferenced to the World Geodetic System 1984 (WGS84) and intersected with the highest level sub-watershed polygons defined by the HYDRO1K global hydrological dataset (U.S. Geological Survey, 2000). This enabled assigning the fractional cover for each land cover type, and hence the dominant land cover type, to the areas immediately surrounding each sampling location.   Swamp includes both temporally and permanently inundated areas of "forest", with vegetation adapted to poorly drained, anaerobic soils (Mayaux et al., 2002).   For all three land cover types the mean annual temperature range (period 1990-2012) is ~1-3°C, temperatures being lowest (~22-24°C) in July-August and highest (~25-26°C) in March-April (http://sdwebx.worldbank.org/climateportal/).   For savannah the average monthly rainfall during July-May (1990-2012) is ~120-260 mm, typically being maximal in October-November, but < 40 mm falls during June-August (http://sdwebx.worldbank.org/climateportal/).   For forest and swamp the annual range in monthly rainfall is less pronounced. Both have two discernable rainfall maxima, during April-May and October-November (~150-240 mm month$^{-1}$), and a minimum in June-August (~40-120 mm month$^{-1}$) (http://sdwebx.worldbank.org/climateportal/).

ROC swamp and forest (Fig. 1) broadly correspond to the westernmost part of the "Cuvette Centrale" (Central Basin). This is a large shallow depression composed mainly of dense, humid forest and extending from approximately 15°W to 25°W and 5°N to 4°S, the central western part of which remains flooded throughout the rainy seasons. Rivers sampled in this region (Sangha, Likouala-aux-Herbes, Likouala, Lengoue, Mambili: Fig. 1, Table 1) drain predominantly sandy or clayey quaternary deposits. The Kouyou basin (Fig. 1) borders the 'Batéké Plateaux', a 600-700m relief sandstone formation to the south, intersected by dry valleys and covering much of the southern ROC. Here, bushy savannah is intersected by the Alima, Nkéni and Léfini rivers. Due to water storage in an underlying sandy-sandstone aquifer the hydrological regimes of these three rivers are largely independent of rainfall; they all show only weak seasonality in discharge despite the relatively large variation in monthly precipitation (Laraque et al., 2001).

## 3 Sample collection and analytical techniques

We collected 66 surface water samples (~0.2 m) from central river channels for dissolved $CH_4$, $N_2O$, $O_2$, $NO_3^-$ and $NH_4^+$ analysis, during November 2010 (41 samples) and August 2011 (25 samples; $CH_4$ and $N_2O$ only). Based on the





monthly rainfall distribution, for convenience we hereinafter refer to these as "wet season" and "dry season" respectively. Samples were slowly decanted into a series of 125 ml glass screw top septum bottles (Sigma-Aldrich, UK) via a silicon rubber tube, over filling each by at least one sample volume to avoid bubble entrainment. Samples were inoculated with 25 μl 0.1 M $HgCl_2$ to arrest microbial activity, sealed to leave no headspace and subsequently

returned to Newcastle for dissolved gas analysis within several weeks of collection. Dissolved gas samples treated in this way can be successfully stored for several months (Elkins, 1980).

Dissolved $CH_4$ and $N_2O$ were analysed by single phase equilibration gas chromatography (Shimadzu GC 14-B), with flame ionisation detection of $CH_4$ and electron capture detection of $N_2O$ (Upstill-Goddard et al., 1996). Routine

calibration was with a mixed secondary standard (361 ppbv $N_2O$, 2000 ppbv $CH_4$) prepared by pressure dilution with ultra-high purity $N_2$ (Upstill-Goddard et al., 1990). Absolute calibration was against a mixed primary standard (10ppmv $N_2O$, 5ppmv $CH_4$) with a certified accuracy of ± 1 % (BOC Special Gases, UK). Overall analytical precisions ($1\sigma$) for $N_2O$ and $CH_4$, established via multiple analysis ($n = 15$) of the secondary standard, were both ± 1%.


Temperature, dissolved $O_2$ and atmospheric pressure were measured *in-situ* using a handheld multi-parameter probe (YSI Pro-Plus, YSI UK Ltd). Quoted measurement accuracies are: ± 0.2 ºC; ± 2 % dissolved $O_2$; ± 0.002 bar. Samples for dissolved $NH_4^+$ and $NO_3^-$ were filtered on collection (Whatman 0.7 μm GF/F; precombusted at 550 °C for 8 h), directly into clean glass vials and stored acidified (pH 2) at 4 °C in the dark for several weeks prior to analysis by

segmented flow (Astoria Analyzer; Astoria-Pacific, USA) at Woods Hole, using established methods (U.S. Environmental Protection Agency, 1984). Analytical precisions ($1\sigma$) were ± 1% for both. Technical and logistical issues precluded the collection of any dissolved $O_2$, $NH_4^+$ or $NO_3^-$ data during August 2011 (dry season) and some $NO_3^-$ and $NH_4^+$ data during November 2010 (wet season).

Emission fluxes, F (mol m$^{-2}$ d$^{-1}$), of $CH_4$ and $N_2O$ were estimated using $F = k_w L\Delta p$, where $k_w$ is the transfer velocity of $CH_4$ or $N_2O$ (cm hr$^{-1}$), $L$ is the solubility of $CH_4$ or $N_2O$ (mol cm$^{-3}$ atm$^{-1}$) (Wiesenburg and Guinasso, 1979; Weiss and Price, 1980) and $\Delta p$ is the corresponding water-to-air partial pressure difference. $k_w$ values were derived from two corresponding estimates for $CO_2$ in the Congo. Raymond et al. (2013) estimated a basin-wide $k_w$ of 5.2 m d$^{-1}$ for $CO_2$, using hydraulic equations involving basin slope and flow velocity. The uncertainty in this estimate is ∼ ± 10%

(Raymond et al., 2013). In contrast Aufdenkampe et al. (2011) applied constant $k_w$ values for $CO_2$ in streams (< 100 m wide: 3.0 m d$^{-1}$) and in rivers (>100m wide: 4.2 m d$^{-1}$). Adjusting for the relative areas of these in the Congo basin (Borges et al., 2015b) gives a basin-wide mean $k_w$ ∼3.9 m d$^{-1}$ for $CO_2$. We converted these estimates to $k_w$ for $CH_4$ and $N_2O$ by multiplying by ($Sc$/470.7)$^{-0.5}$, where 470.7 is the Schmidt number of $CO_2$ in freshwater, and $Sc$ is the Schmidt number of $CH_4$ or $N_2O$ ($Sc_{CH4}$=486.8; $Sc_{N2O}$=476.9), assuming an ambient temperature of 25ºC (Wanninkhof,

1992). The resulting $k_w$ estimates are 5.1 and 3.9 m d$^{-1}$ for $CH_4$ and 5.2 and 4.0 m d$^{-1}$ for $N_2O$. Resulting emissions estimates are consequently ∼30% higher based on Raymond et al. (2013). Using both sets of $k_w$ estimates facilitates a direct comparison with the largest study of $CH_4$ and $N_2O$ fluxes for African rivers that also used this approach (Borges et al., 2015b). While other relevant work used wind based $k_w$ estimates (Koné et al., 2010; Bouillon et al., 2012) the unavailability of wind speeds precludes their use here. We applied the global mean tropospheric mixing ratios of $CH_4$

(1797 ppbv) and $N_2O$ (323 ppbv) for the year 2010 (http://www.eea.europa.eu/data-and-maps/).

### 3 Results

While $CH_4$ and $N_2O$ data are available for all samples, dry season data are not available for dissolved $O_2$, $NO_3^-$ or





$NH_4^+$ and wet season DIN ($NO_3^-$ + $NH_4^+$) is only reported for samples for which both $NO_3^-$ and $NH_4^+$ are available.

### 3.1 Dissolved $O_2$ and DIN

In wet season swamp samples dissolved $O_2$ varied between mildly under-saturated and very strongly under-saturated (Fig. 2). The mean (36 ± 29%) and range (4-91 %) of $O_2$ saturation were both significantly lower than for forest rivers (Mann-Whitney, one-tailed; $P$ = 0.0001), the majority of which were mildly to strongly $O_2$ under-saturated (mean 77 ± 36%, range 14-116 %; Fig. 2), and for savannah rivers (Mann-Whitney, one-tailed; $P$ = 0.002), which were mildly under-saturated to mildly super-saturated (mean 100 ± 17%, range 70-135 %; Fig. 2).


Low wet season concentrations of dissolved inorganic nitrogen (DIN) components (Fig. 3) are consistent with low nitrogen input rates from agricultural, domestic and industrial sources (Clark and Decalo, 2012). The means and ranges of total DIN ($NO_3^-$ + $NH_4^+$) did not differ significantly between any of the three river "types" (mean savannah 6.8 ± 2.8 $\mu$mol $l^{-1}$; range 2.5-10.1 $\mu$mol $l^{-1}$; n=10; mean swamp 5.1 ± 3.1 $\mu$mol $l^{-1}$, range 1.5-10.2 $\mu$mol $l^{-1}$; n= 11;

mean forest 9.4 ± 11.2.$\mu$mol $l^{-1}$, range 1.8-45.3 $\mu$mol $l^{-1}$; n=12), in contrast to the situation for dissolved $O_2$. Differences in $NO_3^-$-N were also not significant (mean savannah 4.1 ± 2.3 $\mu$mol $l^{-1}$, range 0.8-6.7 $\mu$mol $l^{-1}$, n=11; mean swamp 3.6 ± 2.1 $\mu$mol $l^{-1}$, range 1.0-8.8 $\mu$mol $l^{-1}$, n= 16; mean forest 7.1 ± 9.2 $\mu$mol $l^{-1}$, range 1.2-35.1 $\mu$mol $l^{-1}$; n=12) and there was no clear relationship between $NO_3^-$ and $NH_4^+$ for any of the three river types (Fig. 3a). $NO_3^-$ was the dominant DIN component in 24 of the 33 samples for which both $NO_3^-$ and $NH_4^+$ were analysed. Considering all

samples, the mean $NO_3^-$ contribution to DIN was 63 ± 19%.

### 3.2 Dissolved $CH_4$ and $N_2O$

Table 2 summarises ranges, means and medians of riverine $CH_4$ and $N_2O$ concentrations and percent saturations for the three land cover types. All samples were highly $CH_4$ super-saturated, concentrations spanning two orders of magnitude (11.2 - 9553 nmol $L^{-1}$; 440-354400% saturation). $N_2O$ spanned a much narrower concentration range and

varied from strong under-saturation to strong super-saturation (3.2-20.6 nmol $L^{-1}$; 47-205%). Evidently, while rivers of the ROC are strong local sources of tropospheric $CH_4$ they can act as both small sources and sinks for $N_2O$. Swamp rivers exhibited both the lowest and among the highest $N_2O$ saturations (Table 2) but during the wet season had overall significantly lower $N_2O$ than either forest or savannah rivers, which were both always super-saturated (103-266%; Table 2) and for which the overall means and ranges of $N_2O$ were not significantly different (Mann-

Whitney, one-tailed: swamp vs forest and swamp vs savannah, $P$ = 0:004). For $CH_4$, concentration means and ranges during the wet season did not differ significantly between swamp and forest rivers but they were significantly higher in both than in savannah rivers (Mann-Whitney, one-tailed: swamp vs savannah, $P$ = 0:004; forest vs savannah, $P$ = 0.03). In contrast, during the dry season concentration means and ranges of both $CH_4$ and $N_2O$ were indistinguishable for all three land cover types. Seasonal differences in concentration means and ranges were not

significant for $N_2O$ for any of the three land cover types or for $CH_4$ in savannah rivers, but in both swamp and forest rivers $CH_4$ was significantly higher during the wet season (Mann-Whitney, one-tailed: swamp $P$= 0.01; forest $P$ = 0.003).

There are comparatively few measurements of $CH_4$ concentrations in African rivers and even fewer of $N_2O$. Our $CH_4$

data for rivers of the ROC (Table 2) are within the ranges compiled for temperate and tropical rivers (~260-128000%) (Upstill-Goddard et al. 2000; Middelburg et al. 2002) and our $CH_4$ and $N_2O$ data both fall within the ranges recently reported for other rivers in sub-Saharan Africa. Studies of $CH_4$ alone reported 48 - 870 nmol $l^{-1}$ (2221 - 38719 % saturation) in three Ivory Coast rivers (Koné et al., 2010), 25 - 505 nmol $l^{-1}$ (850 - 21700 % saturation) in the Tana



river, Kenya, (Bouillon et al., 2009) and 22-71430 nmol l$^{-1}$ in the Congo (Borges et al., 2015a).  For concurrent measurements of CH$_4$ and N$_2$O Bouillon et al. (2012) report 74 - 280 nmol CH$_4$ l$^{-1}$ (3450 – 13200% saturation) and 6.2 - 9.6 nmol N$_2$O l$^{-1}$ (112-165% saturation) in the Oubangui, a major Congo tributary, Teodoru et al. (2015) found 7-12127 nmol CH$_4$ l$^{-1}$ and 2.0-11.4 nmol N$_2$O l$^{-1}$ at stations along the Zambezi and Borges et al. (2015b) quote a range of 2-62966 nmol CH$_4$ l$^{-1}$ (mean: 2205 nmol l$^{-1}$) and 0.2-85.4 nmol N$_2$O l$^{-1}$ (mean: 9.2 nmol l$^{-1}$) across 12 sub-Saharan river basins, including those of the Congo, Zambezi and Niger.


Considering the complete data set, CH$_4$ was inversely correlated with both N$_2$O and O$_2$ (Fig. 2).  Highest CH$_4$ coincident with lowest N$_2$O and O$_2$ occurred in swamp rivers and lowest CH$_4$ coincident with highest N$_2$O and O$_2$ was observed in forest rivers, while savannah rivers were intermediate between the two (Fig. 2).  Overall, log % CH$_4$ vs % O$_2$ showed a weak negative correlation (Fig. 2a; R$^2$ = 0.26, n = 41) while % N$_2$O vs % O$_2$ showed a weak

positive correlation (Fig. 2b; R$^2$ = 0.30, n = 41). However, for both swamp and forest rivers individually the negative correlations between log % CH$_4$ and % O$_2$ were stronger (swamp R$^2$ = 0.38, n = 16; forest R$^2$ = 0.45, n = 13) and while there was a stronger positive correlation between % N$_2$O and % O$_2$ for swamp rivers than for the complete data set (R$^2$ = 0.71, n = 16), the correlation for forest rivers was extremely weak (R$^2$ = 0.02, n=13).  Conversely, for savannah rivers we found a positive correlation between log % CH$_4$ and % O$_2$ (R$^2$ = 0.23; n= 12) and a negative

correlation between % N$_2$O and % O$_2$ (R$^2$ =0.35, n=12).   N$_2$O co-varied positively with both NO$_3^-$ and NH$_4^+$ (Fig. 3). For the complete data set the correlations were weak (N$_2$O vs NO$_3^-$, R$^2$ = 0.28, n = 59; N$_2$O vs NH$_4^+$, R$^2$ = 0.23, n = 40) but for all three river types individually, with the exception of N$_2$O vs NH$_4^+$ in savannah rivers, the correlations were stronger and for all NO$_3^-$ was a stronger predictor of N$_2$O (N$_2$O vs NO$_3^-$: R$^2$ swamp = 0.50, n = 29;   R$^2$ forest = 0.75, n = 15; R$^2$ savannah = 0.31, n = 15) than was NH$_4^+$ (N$_2$O vs NH$_4^+$: R$^2$ swamp = 0.29, n = 13;   R$^2$ forest = 0.47, n = 13;

R$^2$ savannah = 0.01, n = 14).

### 3.3 CH$_4$ and N$_2$O emission fluxes

Table 3 summarises ranges, means and medians of CH$_4$ and N$_2$O emission fluxes using $k_w$ derived from Raymond et al. (2013) and Aufdenkampe et al. (2011).  Fluxes broadly followed the distribution of concentrations, for CH$_4$ being lowest overall in savannah rivers and highest in swamp and forest rivers and for N$_2$O being lowest in swamp rivers

and highest in savannah and forest rivers.   Fluxes were always to air at all sites for CH$_4$ and at all savannah and forest sites for N$_2$O.   However, swamp rivers were predominantly a N$_2$O sink during the wet season (11 of 16 individual flux estimates) and predominantly a N$_2$O source during the dry season (10 of 16 individual flux estimates).   As far as we are aware the wet season sink for N$_2$O in swamp rivers is the first such reported for African rivers.

### 4 Discussion

### 4.1. Sources of CH$_4$ and N$_2$O

The concentrations of dissolved CH$_4$ and N$_2$O at any specified river location reflect a dynamic and complex balance of in situ production and consumption impacted by import and export mechanisms that include upstream and downstream advection, groundwater inputs, local surface runoff and water-air exchange.

Notwithstanding this complexity, the coexistence of CH$_4$ with dissolved O$_2$ in rivers of the ROC (Fig. 2a) initially seems enigmatic. While dissolved O$_2$ was under-saturated in the majority of samples, being as low as 4% in one wet season swamp sample, it was always detectable and indeed was super-saturated in several savannah river samples in





which $CH_4$ saturations ranged from ~4000-10000 % (Fig. 2a). These observations seem counterintuitive because the classical view of methanogenesis is that it is exclusively anoxic, carried out by severely $O_2$-limited archaea (Bridgham

et al., 2013).    However, recent evidence is for a greater complexity of $CH_4$ production in river catchments. For example, methanogenesis in "anoxic microsites" within oxic soils is widely acknowledged (e.g. Teh et al., 2005; von Fisher & Hedin, 2007). Methanogens are now considered to be widespread in oxic soils and they are activated during flooding (Bridgham et al., 2013), their activity relating to soil carbon age and composition (Bridgham et al., 1998; Chanton et al., 2008) and likely involving substrate competition and other interactions. Production by soil macrofauna

(Kammann et al., 2009), archeal production related to plant productivity (Updegraff et al., 2001; Dorodnikov et al., 2011) and non-microbial, direct aerobic production, both by living plant tissue (Keppler et al., 2006; 2009) and in soils (Hurkuck et al., 2012, have all also been observed. Further, methanogenesis by photoautotroph-attached archaea has been detected in oxic lake water (Grossart et al., 2011), analogous to the "anoxic micro-niches" associated with dead and living particles in oxic sea water (de Angelis and Lee, 1994; Oremland, 1979; Ditchfield et al., 2012).

Additional production in oxic seawater may involve biological uptake of organic $PO_4^{3-}$ (Karl et al., 2008) and methylotrophic methanogenesis (Damm et al., 2010), both mechanisms being associated with nutrient stress, but neither has yet been identified in freshwaters. Additional to this variability in production mechanisms and rates, $CH_4$ is subject to variable and rapid aerobic and anaerobic microbial oxidation (Megonigal et al., 2004); $CH_4$ loss rates have been variously estimated at between a few percent and >100% of the rate of methanogenesis (Bussmann, 2013;

Shelley et al., 2015). Despite such potentially high losses, water to air exchange by ebullition and by turbulent diffusion driven by wind stress, water depth and flow velocity (Raymond and Cole, 2001) is usually considered the major $CH_4$ loss term, with ebullition frequently considered the dominant of these two mechanisms (Stanley et al., 2016).  Despite this complexity of dissolved $CH_4$ cycling in rivers, it is nevertheless informative to speculate on our principal observations in the context of potential $CH_4$ sources and sinks.


The first notable feature of our results is the contrasting relationship between $CH_4$ and $O_2$ in swamp and forest rivers (negative) and in savannah rivers (positive) (Fig. 2a). Dissolved $O_2$ in rivers is primarily driven by the balance between photosynthesis and respiration (Houser et al., 2015) but may also be impacted by varying contributions from water-air exchange that under conditions of extreme turbulence may lead to supersaturations as high as 150% (Li et al.,

2010).    The overall positive relationship between $CH_4$ and $O_2$ in savannah rivers (Fig 2a.) could, at least in part, reflect high macrophyte-related productivity, which can give rise to positive relationships by direct $CH_4$ production (Stanley et al., 2016) and by indirect production via trapping fine-grained organic sediments that support methanogenesis (Sanders et al., 2007). Similar relationships were observed in Amazon floodplain lakes (Devol et al., 1990). Offsetting this, stems and roots respire $O_2$ (Caraco et al., 2006). Further inspection of the data shows that the

highest dissolved $O_2$ saturation found in savannah rivers (134%) deviates from the general $CH_4$ vs $O_2$ trend (Fig. 2a). This sample was collected close to an area of rapids in the Congo main stem, in the vicinity of Stanley Pool (Fig. 1) where other samples were also $O_2$ super-saturated.   Intense water-air exchange in this region via increased turbulence would tend to enhance dissolved $O_2$ (Li et al., 2010) while depleting dissolved $CH_4$. To summarise, notwithstanding possible additional $CH_4$ losses via oxidation, the $CH_4$ vs $O_2$ relationship in savannah rivers (Fig. 2a) could be

explained by net macrophyte production imprinted by water-air gas exchange.   The inverse of this relationship for swamp and forest rivers (Fig. 2a), was similarly reported for the Zambezi and Amazon Basins, for the latter in fast flowing waters (Teodoru et al., 2015; Richey et al., 1988; Devol et al., 1990).   Again, high gas exchange rates are plausible, especially for the small number of tropical forest samples for which $O_2$ was close to or in excess of 100% (Fig. 2a). For the majority of samples that were $O_2$ under-saturated however, additional mechanisms must be invoked.

One possibility is that these distributions largely reflect the mixing of relatively well-oxygenated river waters with



high CH$_4$, low O$_2$ groundwater but another possibility is that this relationship is the aggregate of this and several of the other processes previously discussed.

A second important aspect of the overall CH$_4$ distributions is that swamp and forest river CH$_4$ was highest during the wet season, whereas savannah samples revealed no such inter-seasonal contrast (Table 2). The constancy of CH$_4$ in savannah rivers might well reflect the buffering of seasonal river discharge by the sandy-sandstone aquifer that underlies this region (Laraque et al., 2001). For swamp and forest rivers a number of alternative but not mutually exclusive possibilities might be invoked. In addition to direct and indirect macrophyte production (Stanley et al., 2016; Sanders et al., 2007), as discussed for savannah rivers, methanogenesis following the activation of archaea during the flooding of adjacent soils (Bridgham et al., 2013) is also plausible, especially given that swamp and forest soils are comparatively poorly drained (Mayaux et al., 2002). In contrast, an opposing behaviour was reported for three rivers of the Ivory Coast (Comoé, Bia, Tanoé). In these, overall decreases in CH$_4$ during the dry to wet season transition (Koné et al., 2010) were similar to trends recorded in some temperate (European) rivers (Middelburg et al. 2002). Koné et al. (2010) ascribed the CH$_4$ seasonality in Ivory Coast rivers to a combination of the dilution of high CH$_4$ baseflow by low CH$_4$ surface runoff (e.g. Jones and Mulholland 1998a, b), higher degassing rates during flooding (Hope et al. 2001) and/or decreased in-stream methanogenesis towards high discharge (de Angelis and Scranton, 1993). Conversely, Bouillon et al. (2012) attributed relatively stable high discharge CH$_4$ concentrations (~100 nmol l$^{-1}$) in the Oubangui, a major tributary of the Congo, to terrestrial soil production in conjunction with baseflow transport. The largest fractional CH$_4$ contribution from baseflow often occurs in high elevation headwaters with high soil organic content, while progressive downstream increases in CH$_4$ in lowland rivers have been linked to increasing in-stream methanogenesis (Jones and Mulholland 1998a). Assuming such processes are also operative in ROC swamp and tropical forest, interpreting or predicting the direction of any seasonal CH$_4$ trend in a specified river system is evidently complex.

In contrast to CH$_4$, natural sources of aquatic N$_2$O are entirely microbial, and involve several pathways. Nitrification is a two-stage process in which NH$_4^+$ is first oxidised aerobically to NO$_2^-$ via hydroxylamine (NH$_2$OH), followed by NO$_2^-$ oxidation to NO$_3$. Following the first stage, N$_2$O can be produced through various routes: nitrifier nitrification (NH$_2$OH → N$_2$O), nitrifier denitrification (NO$_2^-$ → NO → N$_2$O) and nitrification-coupled denitrification (NO$_3^-$ → NO$_2^-$ → NO → N$_2$O) (Kool et al., 2011). Heterotrophic denitrification, in which NO$_3^-$ is the terminal electron acceptor (NO$_3^-$ → NO$_2^-$ → NO + N$_2$O → N$_2$), occurs in soils, sediments and water that are anoxic, the inhibition of denitrifier activity at very low levels of dissolved O$_2$ being well known (Knowles 1982). Even so, in the complete absence of O$_2$, N$_2$O can be enzymatically reduced to gaseous N$_2$ (Wrage et al. 2001), both in sediments and in the water column, sometimes resulting in extreme N$_2$O under-saturations (Nirmal Rajkumar et al., 2008).

Although we found no statistically significant differences in the means and ranges of wet or dry season N$_2$O concentrations for any land cover type, higher N$_2$O concentrations and emissions are considered likely where soil-water filled pore spaces exceed 60 % due to enhanced microbial production (Davidson, 1993), as has been observed in African savanna during the rainy season (Castaldi et al., 2006) and throughout much of the year in humid tropical forests (Castaldi et al., 2013). The discrepancy between these and our observations to some extent likely reflects a complex balance between the principal sites (groundwater and in-stream) and mechanisms of N$_2$O cycling, as evidenced by the variable relationships between N$_2$O, O$_2$ and DIN we observed. For example, we found both positive and negative relationships between N$_2$O and O$_2$ (Fig. 2b). Sediment processes and water concentrations are evidently closely coupled in tropical catchments (Harrison and Matson, 2003) and the strong positive correlation



between N$_2$O and O$_2$ in swamp rivers coincident with strong under-saturation of both N$_2$O and O$_2$ (Fig. 2b) is
consistent with N$_2$O consumption by sediment denitrification. Although positive relationships between N$_2$O and
NO$_3^-$ have been variously interpreted to reflect nitrification (Silvennoinen et al., 2008; Beaulieu et al., 2010), or both
denitrification and nitrification (Baulch et al., 2011), for swamp rivers the stronger correlation between N$_2$O and NO$_3^-$
than between N$_2$O and NH$_4^+$, which has previously been taken to indicate a sediment N$_2$O source from denitrification
(Dong et al., 2004), supports our conclusion of a swamp river denitrification sink for N$_2$O. Similar N$_2$O vs O$_2$
relationships were identified in the Amazon and Zambezi river basins (Richey et al., 1988; Teodoru et al., 2015) and
in the Adyar river-estuary, S.E. India (Nirmal Rajkumar et al., 2008). In both the Amazon and the Adyar, N$_2$O was
undetectable in fully anoxic waters (Richey et al., 1998; Nirmal Rajkumar et al., 2008). N$_2$O and NO$_3^-$ were also
correlated in the Oubangui (Bouillon et al., 2012) and a similar, persistent correlation in a temperate river was
ascribed to denitrification in hypoxic/anoxic sediment, favoured by the ambient low river flow and high temperatures
leading to high community respiration and low O$_2$ solubility (Rosamond et al., 2012). Even though denitrification in
rivers may be limited by low levels of NO$_3^-$ (Garcia-Ruiz et al., 1998) a temperate creek nevertheless was a N$_2$O sink
for combined NO$_2^-$ and NO$_3^-$ concentrations < 2.7 μmol l$^{-1}$ (Baulch et al., 2011), broadly similar to the majority of
NO$_3^-$ concentrations we observed (Fig. 3a). By contrast, N$_2$O and NO$_3^-$ were uncorrelated in the Zambezi, for which
there was also no correlation of N$_2$O with NH$_4^+$ (Teodoru et al., 2015). For savannah rivers, in which N$_2$O was
always super-saturated (Fig. 2b), a negative correlation between N$_2$O and O$_2$ may indicate N$_2$O production mainly by
nitrification, a conclusion supported by the corresponding stronger correlation between N$_2$O and NH$_4^+$ than between
N$_2$O and NO$_3^-$, the opposite to what we found for swamp rivers. Although published measurements of N$_2$O production
via in-stream nitrification are lacking, nitrification rates may frequently exceed denitrification rates in streams and
rivers (Richardson et al., 2004; Arango et al., 2008) and nitrification rates are estimated to exceed denitrification rates
two-fold globally (Mosier et al., 1998). In addition to O$_2$ and DIN amount and speciation, pH and dissolved organic
carbon are important in controlling net N$_2$O production via nitrification and denitrification (Baulch et al., 2011) and it
has been suggested that due to variable N$_2$O yields from these processes, simple diagnostic relationships for N$_2$O
production in rivers may prove elusive (Beaulieu et al., 2008).

To conclude, while our data have allowed us to draw some conclusions regarding the production and cycling of CH$_4$
and N$_2$O in contrasting rivers of the ROC, we consider these to be more robust for N$_2$O given that its aquatic sources
are the least diverse. However, for both gases an unequivocal identification of the primary controls of their riverine
distributions would require additional detailed measurements.

**4.2 CH$_4$ and N$_2$O emissions in the wider context**

As with the concentration measurements, there are few data for African rivers with which to compare our CH$_4$ and
N$_2$O emissions estimates (Table 3). Previously published emissions estimates are listed in Table 4. For three rivers
of the Ivory Coast Koné et al. (2010) report 25 - 1187 μmol CH$_4$ m$^{-2}$ d$^{-1}$, while for the Oubangui Bouillon et al. (2012)
found 38 - 350 μmol CH$_4$ m$^{-2}$ d$^{-1}$ and 0.6 - 5.7 μmol N$_2$O m$^{-2}$ d$^{-1}$. For 12 sub-Saharan African rivers Borges et al.
(2015b) give ranges of 502 - 18019 μmol CH$_4$ m$^{-2}$ d$^{-1}$ and 2 - 16 μmol N$_2$O m$^{-2}$ d$^{-1}$ using $k_w$ from Aufdenkampe et al.
(2011), and 583 - 28579 μmol CH$_4$ m$^{-2}$ d$^{-1}$ and 2 - 28 μmol N$_2$O m$^{-2}$ d$^{-1}$ using $k_w$ from Raymond et al. (2013). For
comparison, CH$_4$ emissions estimated for the Amazon River were 4625 - 12562 μmol m$^{-2}$ d$^{-1}$ (Bartlett et al. 1990) and
the range for CH$_4$ in temperate rivers is ~0 - 22000 μmol m$^{-2}$ d$^{-1}$ (De Angelis and Scranton 1993; Lilley et al. 1996;
Jones and Mulholland 1998a, b; Hope et al. 2001; Abril and Iversen 2002). Guérin et al. (2008) reported N$_2$O
emissions ~0.25-6.0 μmol m$^{-2}$ d$^{-1}$ for the Amazon River and floodplain while Soued et al. (2016) found N$_2$O fluxes in
Canadian boreal rivers to be highly variable across ecosystem types and seasons, ranging from net uptake ~ 3.3 μmol





$m^{-2}$ $d^{-1}$, somewhat lower that the maximum $N_2O$ uptake we observed in swamp rivers (Table 3), to net emissions ~ 4.8 $\mu$mol $m^{-2}$ $d^{-1}$.

The overall ranges of $CH_4$ and $N_2O$ emissions from rivers of the ROC (33-48705 $\mu$mol $CH_4$ $m^{-2}$ $d^{-1}$; 1-67 $\mu$mol $N_2O$ $m^{-2}$ $d^{-1}$) are somewhat wider than these earlier estimates for African and temperate rivers, the maximum values (Table 3) being around twice as high as previously reported. Nevertheless, it should be acknowledged that the use of "basin-wide" values for $k_w$ is a necessity that takes no account of spatial and temporal $k_w$ variability, that our emissions based on $k_w$ derived from Raymond et al. (2013) are 30% higher than those derived from Aufdenkampe et al. (2011) and that other available $k_w$ parameterizations show five-fold variability (Barnes and Upstill-Goddard, 2011). Additionally, we 380 did not measure $CH_4$ ebullition fluxes. Borges et al (2015b) report an average 20% ebullition contribution to total $CH_4$ emissions from the Congo and Zambezi, although their maximum estimates are considerably higher than this, and for some other tropical rivers and lakes ebullition is thought to account for 30-98% of total $CH_4$ emissions (Melack et al., 2004; Bastkviken et al., 2010; Sawakuchi et al., 2014). The uncertainties related to $k_w$ notwithstanding, our emissions estimates for $CH_4$ at least, are therefore probably conservative.

**5 Conclusions**

Our data from the ROC support the growing consensus that river systems in Africa may be disproportionately large contributors to the global freshwater sources of tropospheric $CH_4$ and $N_2O$, as they are for $CO_2$, although the potential for significant sinks lends a note of caution for $N_2O$. Nevertheless, the wide ranges of emissions estimates for $CH_4$ and $N_2O$ now available for African rivers clearly illustrate the difficulty in deriving representative total emissions 390 given both the comparatively small size of the available data set and the various approaches that are typically used to derive these emissions. This applies, not only to African rivers but to tropical rivers in general and indeed to freshwaters globally. At least equally important is an insufficiently mature understanding of the processes that link emissions to the environmental controls of process rates and their temporal variability, and to river catchment characteristics that include sources and seasonality of organic inputs and variability in the balance between baseflow 395 and surface runoff. Our understanding of these interactions must improve if the system responses to future climate and land use changes are to be predicted and planned for. Lastly, the measurement of $CH_4$ and $N_2O$, data calibration and the emissions estimates deriving would all benefit from agreed, standardized protocols. This is an issue that is yet to be adequately addressed, not only for freshwaters but for aquatic systems more generally.

*Acknowledgements.* We appreciate the generous logistical support provided by the Wildlife Conservation Society, 400 Congo (WCS-Congo) and a substantial contribution to field costs from the School of Marine Science and Technology at Newcastle University. Without either of these contributions this study would not have been feasible.

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






**List of Figures**

**List of Tables**





Figure 1.



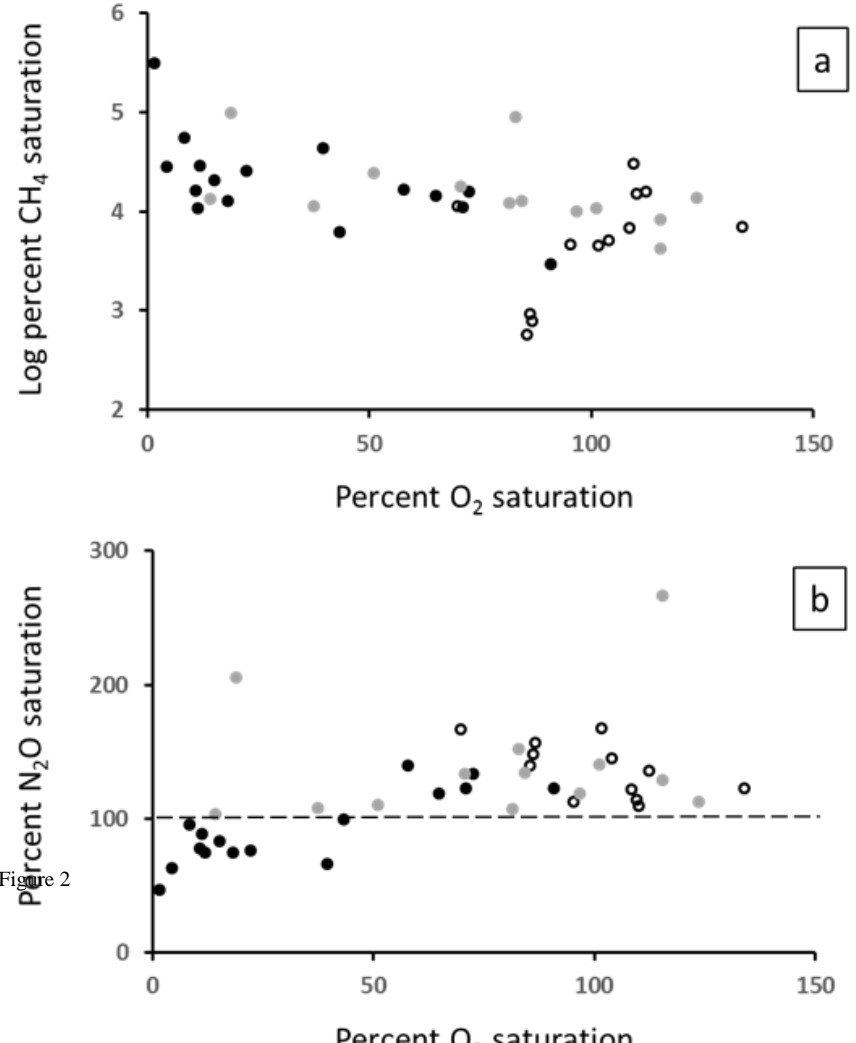

Figure 2





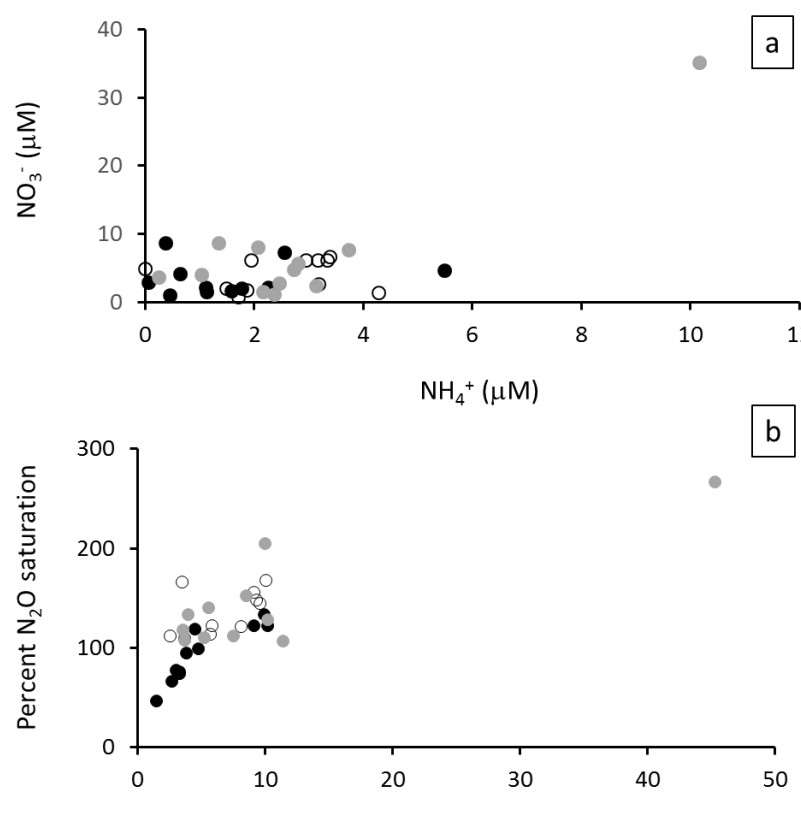

Figure 3



**Table 1.** Relevant physical characteristics of rivers studied in this work. All rainfall data are from Laraque et al. (2001), Djoue catchment area and discharge data are from Laraque et al. (1994) and all other data are from Laraque et al. (2009).

| River/Tributary | | | Catchment Area km$^2$ | Discharge m$^3$ s$^{-1}$ | Rainfall mm |
|---|---|---|---|---|---|
| Congo | | | 3500000 | 40600 | 1528 |
| | Alima | | 21030 | 1941 | 1709 |
| | Nkèni | | 8000 | 261 | 1662 |
| | Léfini | | 14000 | 400 | 1615 |
| | Djoue | | 5740 | 140 | 1547 |
| | Likouala aux-Herbes | | 25000 | 267 | 1622 |
| | | Sangha | 211120 | 1941 | 1511 |
| | Likouala Mossaka | | 69800 | 928 | 1689 |
| | | Kouyou | 16000 | 191 | 1566 |
| | | Lengoué | 12125 | 155 | |
| | | Mambili | 13700 | 161 | |
| | Motaba | | 772800 | 4000 | |





**Table 2.** Dissolved CH$_4$ and N$_2$O in Republic of Congo rivers

| Land cover | Date | CH$_4$ nmol L$^{-1}$ | | | CH$_4$ saturation % | | | N$_2$O nmol L$^{-1}$ | | | N$_2$O saturation % | | |
|---|---|---|---|---|---|---|---|---|---|---|---|---|---|
| | | Range | Mean | Median | Range | Mean | Median | Range | Mean | Median | Range | Mean | Median |
| Savannah | All | 15-793 | 204 ± 195 | 150 | 571-29721 | 7442 ± 73427 | 5444 | 7.6-11.9 | 8.9 ± 1.1 | 8.6 | 6-167 | 131 ± 19 | 122 |
| | Wet | 15-793 | 229 ± 141 | 153 | 571-29721 | 8505 ± 4111 | 5936 | 7.6-11.9 | 9.1 ± 2.2 | 9.0 | 106-167 | 136 ± 34 | 137 |
| | Dry | 66-243 | 144 ± 70 | 150 | 1660-8717 | 4888 ± 2878 | 5444 | 8.0-9.2 | 8.4 ± 0.5 | 8.2 | 113-130 | 119 ± 6 | 117 |
| Swamp | All | 17-9553 | 955 ± 2150 | 299 | 659-354444 | 35025 ± 79578 | 10913 | 3.2-15.3 | 7.2 ± 2.2 | 7.2 | 47-205 | 101 ± 30 | 98 |
| | Wet | 77-8394 | 1052 ± 1992 | 456 | 2945-309533 | 38485 ± 73488 | 16292 | 3.2-10.3 | 6.5 ± 2.1 | 5.7 | 47-139 | 92 ± 28 | 86 |
| | Dry | 17-9553 | 858 ± 2359 | 196 | 659-354444 | 31565 ± 87529 | 7090 | 5.3-15.2 | 7.9 ± 2.2 | 7.7 | 78-205 | 111 ± 29 | 109 |
| Tropical Forest | All | 11-2730 | 551 ± 784 | 321 | 440-98582 | 16966 ± 22958 | 11302 | 7.6-20.6 | 89.6 ± 3.3 | 8.4 | 103-266 | 135 ± 42 | 28 |
| | Wet | 110-2730 | 691 ± 852 | 331 | 4205-98582 | 21205 ± 24858 | 12729 | 7.6-20.6 | 10.0 ± 3.7 | 5.7 | 103-266 | 140 ± 47 | 128 |
| | Dry | 11-232 | 94 ± 103 | 67 | 440-82183 | 3319 ± 3360 | 2310 | 8.0-8.5 | 8.3 ± 0.2 | 8.3 | 108-132 | 120 ± 12 | 120 |

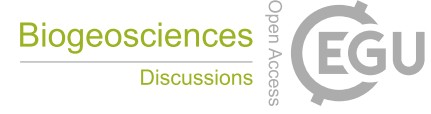

**Table 3.** Emissions of CH$_4$ and N$_2$O to air from rivers in the Republic of Congo. Negative values in parentheses indicate uptake from the atmosphere.

| Land cover | Date | CH$_4$ emission flux ($\mu$mol m$^{-2}$ d$^{-1}$) | | | | | | N$_2$O emission flux ($\mu$mol m$^{-2}$ d$^{-1}$) | | | | | |
| | | Raymond et al. (2013) | | | Aufdenkampe et al. (2011) | | | Raymond et al. (2013) | | | Aufdenkampe et al. (2011) | | |
| | | Range | Mean | Median | Range | Mean | Median | Range | Mean | Median | Range | Mean | Median |
|---|---|---|---|---|---|---|---|---|---|---|---|---|---|
| Savannah | | | | | | | | | | | | | |
| | All | 61-4030 | 1027 ± 997 | 752 | 47-3081 | 785 ± 762 | 575 | 3-25 | 11 ± 7 | 8 | 3-19 | 8 ± 5 | 6 |
| | Wet | 61-4030 | 1159 ± 1157 | 876 | 47-3081 | 887 ± 885 | 670 | 3-25 | 12 ± 7 | 13 | 3-19 | 9 ± 5 | 10 |
| | Dry | 321-855 | 595 ± 356 | 602 | 245-654 | 455 ± 272 | 460 | 5-11 | 7 ± 2 | 6 | 4-8 | 5 ± 2 | 5 |
| Swamp | | | | | | | | | | | | | |
| | All | 74-48705 | 4856 ± 10964 | 1512 | 56-37245 | 3713 ± 8384 | 1196 | (-19)-41 | 1 ± 11 | (-1) | (-15)-31 | 1 ± 9 | (-1) |
| | Wet | 378-42798 | 5349 ± 10160 | 2313 | 289-32728 | 4090 ± 7769 | 1769 | (-19)-15 | (-3) ± 10 | (-5) | (-15)-12 | (-2) ± 8 | (-4) |
| | Dry | 74-48705 | 4363 ± 12029 | 985 | 56-37245 | 3336 ± 9198 | 753 | (-8)-41 | 4 ± 11 | 3 | (-6)-31 | 3 ± 9 | 3 |
| Tropical Forest | | | | | | | | | | | | | |
| | All | 43-13911 | 2795 ± 4000 | 1624 | 33-10638 | 2318 ± 3059 | 1242 | 1-67 | 13 ± 16 | 9 | 1-51 | 10 ± 13 | 7 |
| | Wet | 549-13911 | 3512 ± 4387 | 1677 | 420-10638 | 2686 ± 3324 | 1282 | 1-67 | 15 ± 19 | 10 | 1-51 | 11 ± 14 | 7 |
| | Dry | 43-1167 | 465 ± 526 | 326 | 33-892 | 356 ± 403 | 249 | 3-11 | 7 ± 3 | 7 | 3-8 | 15 ± 3 | 5 |





**Table 4.** Emissions of $CH_4$ and $N_2O$ published for African rivers: (A) refers to emissions estimated using the relationship of Aufdenkampe et al (2011) and (B) refers to emissions estimated using the relationship of Raymond et al. (2013).

| River | CH₄ emission flux ($\mu$mol m$^{-2}$ d$^{-1}$) | | N₂O emission flux ($\mu$mol m$^{-2}$ d$^{-1}$) | | | Reference |
|---|---|---|---|---|---|---|
| | Range | Mean | Range | Mean | | |
| Conoé, Ivory Coast | | 288 ± 107 | | | | Koné et al (2010) |
| Bia, Ivory Coast | | 155 ± 38 | | | | Koné et al (2010) |
| Tanoé, Ivory Coast | | 241 ± 91 | | | | Koné et al (2010) |
| Ivory Coast (all) | 25-1187 | | | | | Koné et al (2010) |
| Oubangui | 38-350 | | 0.6-5.7 | | | Bouillon et al (2012) |
| Congo | | 14296 | | 15 | (A) | |
| | | 18534 | | 19 | (R) | Borges et al (2015b) |
| Ivory Coast | | 1003 | | | (A) | |
| | | 1667 | | | (R) | Borges et al (2015b) |
| Ogooué | | 2115 | | 13 | (A) | |
| | | 4668 | | 28 | (R) | Borges et al (2015b) |
| Niger | | 502 | | 4 | (A) | |
| | | 583 | | 5 | (R) | Borges et al (2015b) |
| Zambezi | | 8348 | | 2 | (A) | |
| | | 13597 | | 2 | (R) | Borges et al (2015b) |
| Betsiboka | | 1305 | | 4 | (A) | |
| | | 3493 | | 9 | (R) | Borges et al (2015b) |
| Rianila | | 1923 | | 5 | (A) | |
| | | 4537 | | 12 | (R) | Borges et al (2015b) |
| Tana | | 568 | | 6 | (A) | |
| | | 604 | | 6 | (R) | Borges et al (2015b) |
| Athi-Galana-Sabaki | | 1156 | | 16 | (A) | |
| | | 1374 | | 19 | (R) | Borges et al (2015b) |
| Nyong | | 18019 | | | (A) | |
| | | 28579 | | | (R) | Borges et al (2015b) |