# Peer review of "The riverine source of CH4 and N2O from the Republic of Congo, Western Congo Basin"

_Biogeosciences, 2016_

## Referee Comment (RC1) · Anonymous Referee #1 · 16 Nov 2016

General comment

Upstill-Goddard and colleagues report a valuable data-set of CH4 and N2O in the Congo basin in the Republic of Congo (ROC).

Major comment

I suggest that the authors make their data-set public as a supplement of the paper. Considering the enormous range of (spatial and temporal) variability of CH4 (and to a lesser extent of N2O) in freshwaters, there is a need to compile and aggregate available data-sets to revise and update CH4 fluxes from inland waters. This is only possible if an open data access attitude is adopted by the community.

[Figure]

Specific comments

L 47 : Alternative explanations have been proposed, such as related to agriculture (Schaefer et al. 2016) or fossil fuel (Rice et al. 2016), that differ from the explanation from Bousquet et al.

L 59 : This is not correct any more in the light of the paper by Stanley et al. 2016.

L 60 : It could be useful to mention that there's a discrepancy in bottom-up and top-down estimates of CH4 fluxes (Saunois et al. 2016), hence the comparison of Bastvinken et al. estimates with those of Kirschke et al. might be biased by the fact that they were derived by different and possibly incompatible methods.

L 171 : This explanation for the seasonal variations of DIN is surprising given these are near pristine watersheds due to the low population density in ROC and the absence of intensive agricultural practise (based on artificial fertilizer) and major industrial activities. Seasonal variations of DIN are likely due to surface run-off in the wet season and groundwater flow in deeper soil horizons in the dry season.

L 240-264: The existence, in rivers and wetlands, of high levels of CH4 in oxygenated waters is not "enigmatic" nor "counterintuitive" as stated. This has been shown and explained for decades for instance in the Amazon (Richey et al. 1988), and is related to methane production in the anoxic sediments of river-beds and floodplains that diffuses into aerated river water. In shallow and low turbulent "swampy" waters such as those sampled, the diffusion of CH4 from river sediments is stronger than loss terms in the water by oxidation or evasion to the atmosphere, leading to an accumulation of CH4 in the water (even in the presence of more or less large quantities of O2). This is fairly straightforward and intuitive, in rivers with a probable depth between 1 and 5 m, hence, in close contact with organic rich sediments. There is no need to use exotic hypothesis related to DMS(P) cycling (Dam) or methylphosphanate (Karl) that were developed for the ocean, where the occurrence of CH4 in oxygenated waters located hundredths to thousands of meters away from the seabed is indeed "enigmatic", hence, the so called

"oceanic CH4 paradox". Further, most of these hypothesis rely on a more or less direct production of CH4 linked to phytoplankton (e.g. Grossart). However, phytoplankton is nearly absent in tributaries and wetlands of the Congo (Descy et al. 2016).

L 276: Please use the term "Pool Malebo", the term "Stanley Pool" has been abandoned since colonial times.

L 289-308 : Seasonal variations of CH4 in floodplains has been relatively well described in the Amazon varzeas (Devol et al. 1990).

L348-349: Rates of nitrification can exceed denitrification in NH4 enriched temperate rivers such as the Mississippi studied by Richardson et al., however this does not necessarily apply in DIN poor tropical rivers where as stated NO3 dominates the DIN pool.

L 357 : If the authors envisage all possible CH4 sources in marine and freshwater environments then CH4 has more diverse sources that N2O. However, the only documented CH4 sources in tropical rivers are methanogenesis in riverbed and floodplain sediments.

L 363-365 : The cited range of CH4 and N2O fluxes correspond to the basin average values and not the full range of individual CH4 and N2O flux estimates for each of the 12 river basins.

The range of CH4 fluxes for all individual estimates across the 12 rivers studied by Borges et al. (2015) is 0 to 274,600 $\mu$mol/m2/d for Aufdenkampe K estimate and 0 to 461,967 $\mu$mol/m2/d for Raymond K estimate. This range in fact corresponds to the one of the Congo that encompasses the data from all other African rivers.

The range of N2O fluxes for all individual estimates across the 12 rivers studied by Borges et al. (2015) is -30 to 299 $\mu$mol/m2/d for Aufdenkampe K estimate and -37 to 377 $\mu$mol/m2/d for Raymond K estimate. This range in fact corresponds to the one of the Congo that encompasses the data from all other African rivers.

L 375 : Hence, statement that the range of CH4 and N2O fluxes from rivers ROC is wider than previously reported for African rives is incorrect and is based on the comparison of individual estimates in ROC to the basin averaged values reported by Borges et al. (2015).

L 382 : I suggest to limit this comparison to rivers that are more turbulent than lakes, hence, diffusion is likely to dominate over ebullition (unlike lakes) due to higher gas transfer velocities and lower settling of organic matter in sediment compared to lakes. In rivers, CH4 diffusion:ebullition data from Congo and Zambezi (Borges et al. 2015) converge with data in the Amazon (Sawakuchi et al. 2014).

L 370 : Is there a point to the comparison with Canadian boreal rivers ?

References

Descy JP, Darchambeau F, Lambert T, Stoyneva MP, Bouillon S, Borges AV (2016) Phytoplankton dynamics in the Congo River, Freshwater Biology, DOI: 10.1111/fwb.12851

Devol, A.H., Richey J.E., Forsberg B.R., Martinelli, L.A. Seasonal dynamics of CH4 emissions from the amazon river floodplain to the troposphere. J. Geophys. Res. 95, 16417-16426 (1990).

Rice et al. (2016) Atmospheric methane isotopic record favors fossil sources flat in 1980s and 1990s with recent increase, PNAS, doi: 10.1073/pnas.1522923113

Richey, J.E., Devol, A.H., Wofy, S.C., Victoria, R. & Riberio, M.N.G. Biogenic gases and the oxidation and reduction of carbon in Amazon River and floodplain waters. Limnol. Oceanogr. 33, 551-561 (1988).

Saunois M et al. (2016) The Global Methane Budget: 2000-2012, Earth Syst. Sci. Data Discuss., doi:10.5194/essd-2016-25

Schaefer et al. (2016) A 21st century shift from fossil-fuel to biogenic methane emissions indicated by 13CH4 Science , DOI: 10.1126/science.aad2705

Sawakuchi, H.O. et al. Methane emissions from Amazonian Rivers and their contribution to the global methane budget. Global Change Biol. 20, 2829-2840 (2014).

Stanley EH, Casson NJ, Christel ST, Crawford JT, Loken LC, Oliver SK. 2016. The ecology of methane in streams and rivers: Patterns, controls, and global significance. Ecological Monographs 86: 146–171.

---

## Referee Comment (RC2) · Anonymous Referee #2 · 2 Dec 2016

The brief paper by Upstill-Goddard presents a dataset on riverine dissolved CH4 and N2O concentrations from a range of rivers and streams in the western Congo Basin, an area from which few data exist. For the wet season sampling, the dataset includes some complementary data such as nitrate and ammonia concentrations, and dissolved O2. While not really providing much novelty in terms of identifying drivers of the CH4 and N2O balance in this type of systems, as the amount and type of ancillary data is rather limited and restricted to the wet season sampling, the dataset is valuable and merits publication – but I strongly agree with Reviewer #1 that it is crucial that the full data are made available in digital form, there is little point in collecting these data and leaving a legacy of just average or mean values in a summary Table.

My comments and suggestions are relatively minor and should be straightforward to incorporate.

-Title: I find the title a little awkward, the 'riverine source of tropospheric CH4 and N2O' is somewhat misleading as the paper does not focus on budgeting sources of tropospheric CH4 and N2O, it looks at CH4 and N2O exchange (source, or sink in some cases for N2O) between surface waters and atmosophere. I suggest to rephrase the title.

-L19-20: 'predominantly supersaturated': with an average of 100% saturation, why call this 'predominantly oversaturated' ?

-L22: The abbreviation ROC is used here for the first time – spell it out here.

-L32-33: awkward sentence, not clear what the authors are trying to say here ('was coincident with')

-L70: what do you mean with 'seasonal emissions' ?

-L124: inoculated: poisoned

-L240-264: I suggest deleting this entire section, not relevant for freshwater systems as pointed out by Reviewer #1.

---

## Short Comment (SC1) · 10 Dec 2016

I would like to add some additional comments regarding the use of references in this work, since overall more care needs to be given that statements are worded accurately:

L41-42: Hartmann et al. 2013 does not describe the impacts of CH4 on oxidizing capacity and other atmospheric species. Myhre et al. 2013 is a suitable reference

L47-48, also L53-54: This is the global surface mean, not the tropospheric mean, which would be slightly lower. Chapter 2 of AR5 (Hartmann et al. 2013) is a better citation than Chapter 6 (Ciais et al. 2013) since it details the data sources where these numbers are derived from. Also, simply say "150%" rather than "more than 150%".

[Figure]

L48-49: It is not clear why "soils, freshwaters and coastal waters" is given. Freshwater and coastal waters were not explicitly discussed in the reference, and this excludes important sources they do give including animal waste management, biomass burning, oceans, and energy systems.

L50-51: This has several issues and does not follow from the previous sentence. 1) rivers, estuaries, and coastal zones was already included as a source category for N2O in IPCC AR4 (see Denman et al. 2007), though the sentence seems to imply it was added in AR5. 2) it is not that they switched the classification from natural to anthropogenic, but that sufficiently reliable data became available on the anthropogenic component.

L60-61: "based on warming potentials and atmospheric lifetimes" -> "based on a 100-year Global Warming Potential" would be more descriptive

L60-62: Kirschke et al. 2013 gives a bottom-up value of 40 [8-73] Tg/yr for fresh-water sources, and a total source of 678 [542-852] Tg/yr, giving a mean freshwater contribution of about ∼6% along with a wide uncertainty range. The estimates given of ∼30-47% for natural and ∼12-20% total sources seem to be miscalculated. In addition, it is unclear why we are asked to ignore the range of 10 − 100 Tg/yr given on L57 for freshwater, which would yield an even larger range of uncertainty

L63: Please double check this. It does not seem to be consistent with Figure 2 in Le Quéré et al. 2015 which suggests a combined sink of ∼5.5 x 10^15 g C yr-1

L159-160: Same comment about being surface rather than tropospheric mean CH4 and N2O. Also unclear how to access the referenced data using the link given at http://www.eea.europa.eu/data-and-maps/

L269: supersaturations -> super-saturations (for consistency with usage in the rest of the text)

L383: Melack et al. 2004 and Bastkviken et al. 2010 both missing from References list

L396-398: It is clear there are issues with emissions estimation that need to be addressed in the future, but it is not clear with the issue is with CH4 and N2O measurement and data calibration. It seems that standard, accurate, techniques are available for collection of samples and subsequently measuring CH4 and N2O concentrations - please remove or clarify this point.

―――――――――――――――――――――

---

## Author Comment (AC1) · 20 Jan 2017

**bg-2016-404:** The riverine source of tropospheric $CH_4$ and $N_2O$ from the Republic of Congo, Western Congo Basin

Response to referee comments:

Three sets of reviewer comments were received for this manuscript. Below we reproduce these comments, italicised, and followed by our responses, which are un-italicised. Text that we have now deleted/modified is highlighted in grey, with text lines in the original manuscript identified. Replacement/modified text is highlighted in yellow, with new manuscript text lines identified. The replacement/modified text is also highlighted in yellow on the revised manuscript.

**Anonymous Reviewer #1**

*I suggest that the authors make their data-set public as a supplement of the paper. Considering the enormous range of (spatial and temporal) variability of CH4 (and to a lesser extent of N2O) in freshwaters, there is a need to compile and aggregate available data-sets to revise and update CH4 fluxes from inland waters. This is only possible if an open data access attitude is adopted by the community.*

We absolutely agree; however, we did not want to make our data public at the initial review stage, preferring to await full publication of the manuscript. We now include a supplementary data file that list all relevant data.   We have included the source dat as supplementary Table S1 and we now include the statement: "Source data for this paper are available as supplementary material (Table S1)", in new text line 173.

*L47: Alternative explanations have been proposed, such as related to agriculture (Schaefer et al. 2016) or fossil fuel (Rice et al. 2016), that differ from the explanation from Bousquet et al.*

While this is true, very recent isotopic work (*Nisbet et al.*  Rising atmospheric methane: 2007–2014 growth and isotopic shift. *Glob. Biogeochem. Cyc*. 30, 1356–1370, doi:10.1002/2016GB005406, 2016) now provides strong evidence for recently increased biogenic emissions, especially in the tropics, where these may reflect an expansion of tropical wetlands in response to positive rainfall anomalies.  In view of this new evidence, we have removed the original text in lines 44-47: "Following periods of declining and zero growth from the mid-1980s, tropospheric $CH_4$ progressively increased from the late 2000s (Rigby et al., 2008; Dlugokencky et al., 2009). Growth ~ 4-5 ppbv $yr^{-1}$ since 2009 (Sussmann et al., 2012) has been linked to increasing natural tropical emissions (Bousquet et al., 2011".  This is now replaced with new text lines 46-52: "The early 1980s to the mid-2000s saw an overall decline in tropospheric $CH_4$ growth (Dlugokencky et al., 2009; 2011), which has been ascribed to declining fossil fuel emissions and/or variations in the OH radical sink (Rice et al., 2016; Schaefer et al., 2016), punctuated by episodic events such as the 1991-1992 Pinatubo eruption and an intense 1997–1998 El Niño (Nisbet et al., 2016). However, since 2007 increased growth has been sustained.  Recent evidence from isotopic studies (Nisbet et al., 2016) and a box-model (Schaefer et al., 2016) implies this to be a consequence of increased biogenic emissions, particularly in the tropics, where these increased emissions have been linked to an expansion of tropical wetlands in response to positive rainfall anomalies (Nisbet et al., 2016), and/or growing emissions from agricultural sources (Schaefer et al., 2016).

References to Bousquet et al. (2011), Rigby et al. (2008) and Sussman et al. (2012) are now redundant and have been removed from the reference list. Nisbet et al. (2016), Dlugokencky et al. (2011), Rice et al. (2016) and Schaefer et al. (2016) have been added.

*L 59: This is not correct any more in the light of the paper by Stanley et al. 2016.*

*L 60 : It could be useful to mention that there's a discrepancy in bottom-up and top-down estimates of CH4 fluxes (Saunois et al. 2016), hence the comparison of Bastvinken et al. estimates with those of Kirschke et al. might be biased by the fact that they were derived by different and possibly incompatible methods.*

We considered these two comments together because they are linked in the text. In light of these we have revised the original text in 57-59: "The global freshwater $CH_4$ source could be ~$10^{13}$-$10^{14}$ g $yr^{-1}$ (Bastviken et al., 2011; Kirschke et al., 2013), the uncertainty reflecting data gaps, notably for major world river basins, and a sampling bias that has necessitated upscaling from exclusively temperate data (Bastviken et al., 2011)". Noting that the estimate of Stanley et al (2016) falls within the range set by Bastviken et al. (2011) and Kirschke et al. (2013), the new text reads (new text lines 61-66) now reads: "The global freshwater $CH_4$ source could be ~$10^{13}$-$10^{14}$ g $yr^{-1}$ (Bastviken et al., 2011; Kirschke et al., 2013; Stanley et al., 2016). This order of magnitude range largely reflects a discrepancy between "top-down" approaches based on atmospheric inversions (e.g. Kirschke et al., 2013) and "bottom-up" estimates that necessitate the upscaling of freshwater observations (e.g. Bastviken et al., 2011). For example, atmospheric constraints on top-down budgets imply that some component emissions of bottom-up approaches may be overestimates (Saunois et al., 2016)".

We have added the Saunois et al. (2016) to the reference list.

*L 171 : This explanation for the seasonal variations of DIN is surprising given these are near pristine watersheds due to the low population density in ROC and the absence of intensive agricultural practise (based on artificial fertilizer) and major industrial activities. Seasonal variations of DIN are likely due to surface run-off in the wet season and groundwater flow in deeper soil horizons in the dry season.*

We find this comment perplexing because we did not comment at all on seasonal variations in DIN. This is because we only have DIN data for the wet season, something that we made absolutely clear in the opening sentence of the results section (original text lines 162-163). We could only compare the three river "types" during the wet season and see no reason therefore, to speculate about wet vs dry season sources of DIN here, given that we have no data for the latter. Perhaps the reviewer does not mean to refer to seasonal variation at all but has simply become a little confused and has misinterpreted what it was we were intending to say in general about the importance of agricultural DIN sources. Our original text, lines 171-172: "Low wet season concentrations of dissolved inorganic nitrogen (DIN) components (Fig. 3) are consistent with low nitrogen input rates from agricultural, domestic and industrial sources (Clark and Decalo, 2012)" means to imply exactly what the reviewer argues, i.e. that agricultural impacts are negligible. To preclude any further misunderstanding however, we have modified this text so that it now reads (new text lines 181-182): "Low wet season concentrations of dissolved inorganic nitrogen (DIN) components (Fig. 3) are consistent with agricultural, domestic and industrial DIN sources all being negligible (Clark and Decalo, 2012)". We trust that this is even clearer than it was previously. In view of this we have also

modified the corresponding text in the abstract: "total DIN concentrations (1.5-45.3 µmol L$^{-1}$) being consistent with small agricultural, domestic and industrial sources". "total DIN concentrations (1.5-45.3 µmol L$^{-1}$) being consistent with negligible agricultural, domestic and industrial sources" (text lines 17-18).

*L 240-264: The existence, in rivers and wetlands, of high levels of CH4 in oxygenated waters is not "enigmatic" nor "counterintuitive" as stated. This has been shown and explained for decades for instance in the Amazon (Richey et al. 1988), and is related to methane production in the anoxic sediments of river-beds and floodplains that diffuses into aerated river water. In shallow and low turbulent "swampy" waters such as those sampled, the diffusion of CH4 from river sediments is stronger than loss terms in the water by oxidation or evasion to the atmosphere, leading to an accumulation of CH4 in the water (even in the presence of more or less large quantities of O2). This is fairly straightforward and intuitive, in rivers with a probable depth between 1 and 5 m, hence, in close contact with organic rich sediments.*

We accept this point completely. We have therefore deleted the following original text (original text lines 240-249): "Notwithstanding this complexity, the coexistence of CH$_4$ with dissolved O$_2$ in rivers of the ROC (Fig. 2a) initially seems enigmatic. While dissolved O$_2$ was under-saturated in the majority of samples, being as low as 4% in one wet season swamp sample, it was always detectable and indeed was super-saturated in several savannah river samples in which CH$_4$ saturations ranged from ~4000-10000 % (Fig. 2a). These observations seem counterintuitive because the classical view of methanogenesis is that it is exclusively anoxic, carried out by severely O2-limited archaea (Bridgham et al., 2013). However, recent evidence is for a greater complexity of CH4 production in river catchments. For example, methanogenesis in "anoxic microsites" within oxic soils is widely acknowledged (e.g. Teh et al., 2005; von Fisher & Hedin, 2007). Methanogens are now considered to be widespread in oxic soils and they are activated during flooding (Bridgham et al., 2013), their activity relating to soil carbon age and composition (Bridgham et al., 1998; Chanton et al., 2008) and likely involving substrate competition and other interactions".

This is now replaced by new text which is informed by the reviewer's comments (new text lines 250-263): "While dissolved O$_2$ was under-saturated in the majority of samples, being as low as 4% in one wet season swamp sample, it was always detectable and indeed was super-saturated in several savannah river samples in which CH$_4$ saturations ranged from ~4000-10000 % (Fig. 2a). Notwithstanding that methanogenesis is an exclusively anoxic process carried out by severely O$_2$-limited archaea (Bridgham et al., 2013), the existence of high CH$_4$ concentrations in oxygenated rivers is well-known (e.g. Richey et al., 1988). It is a consequence of the diffusion of CH$_4$ produced in underlying river sediments and in adjacent floodplain soils into aerated river water. In shallow rivers with low levels of surface turbulence such as those studied here, the CH$_4$ diffusion term evidently exceeds combined CH$_4$ losses via oxidation and water-to-air exchange, resulting in the accumulation of high river water CH$_4$ concentrations.

In addition to methanogenesis in fully anoxic sediment and soils however, CH$_4$ production can also occur in "anoxic microsites" within oxic soils (e.g. Teh et al., 2005; von Fisher & Hedin, 2007). Indeed, methanogens are now considered to be widespread in oxic soils and they are activated during flooding (Bridgham et al., 2013), their activity relating to soil carbon age and composition (Bridgham et al., 1998; Chanton et al., 2008) and likely involving substrate competition and other interactions".

*There is no need to use exotic hypothesis related to DMS(P) cycling (Dam) or methylphosphanate (Karl) that were developed for the ocean, where the occurrence of CH4 in oxygenated waters located hundredths to thousands of meters away from the seabed is indeed "enigmatic", hence, the so called "oceanic CH4 paradox". Further, most of these hypothesis rely on a more or less direct production of CH4 linked to phytoplankton (e.g. Grossart). However, phytoplankton is nearly absent in tributaries and wetlands of the Congo (Descy et al. 2016).*

The inclusion of this was a response to an initial request from an Associate Editor that we discuss ALL aquatic sources of $CH_4$. We fully concur with the reviewer's view and we are happy to have now deleted the offending text (original lines 255-258): "Additional production in oxic seawater may involve biological uptake of organic $PO_4^{3-}$ (Karl et al., 2008) and methylotrophic methanogenesis (Damm et al., 2010), both mechanisms being associated with nutrient stress, but neither has yet been identified in freshwaters". Damm et al (2010) and Karl et al (2008) have been removed from the reference list.

We have also modified the preceding sentence (original lines 252-254): "Further, methanogenesis by photoautotroph-attached archaea has been detected in oxic lake water (Grossart et al., 2011), analogous to the anoxic micro-niches"associated with dead and living particles in oxic sea water (de Angelis and Lee, 1994; Oremland, 1979; Ditchfield et al., 2012)" replacing this by text lines 266-268: "Although methanogenesis by photoautotroph-attached archaea has been detected in oxic lake water (Grossart et al., 2011), this is unlikely in tributaries and wetlands of the Congo, where phytoplankton abundance is low (Descy et al. 2016)". Consequently, de Angelis and Lee (1994), Oremland (1979) and Ditchfield et al. (2012) have been deleted from the reference list and Descy et al (2016) has been added.

*L 276: Please use the term "Pool Malebo", the term "Stanley Pool" has been abandoned since colonial times.*

We note that "Stanley Pool" still appears on some maps and in some relatively recent documents. To provide continuity in this regard we have replaced "in the vicinity Stanley Pool (Fig. 1)" by (new text lines 287-288): "in the vicinity of Pool Malebo (Formerly known as Stanley Pool) (Fig. 1)"

*L 289-308 : Seasonal variations of CH4 in floodplains has been relatively well described in the Amazon varzeas (Devol et al. 1990).*

We are unable to see how this comment refers to the text passage identified or where it could fit in, especially as no supporting argument is presented by the referee. Even so, we think this is a minor point that does not add to the thrust of our reasoning.

*L348-349: Rates of nitrification can exceed denitrification in NH4 enriched temperate rivers such as the Mississippi studied by Richardson et al., however this does not necessarily apply in DIN poor tropical rivers where as stated NO3 dominates the DIN pool.*

Although nitrate does indeed dominate the DIN pool it does not necessarily follow that nitrification would thus be unimportant. As we stated earlier, nitrate accounts for 63 ± 19% of the sum of nitrate

and ammonium so the latter is still important. We are therefore unsure what the referee is asking, especially as he/she uses the phrase "does not necessarily apply". We do not believe this to be a strong case for dismissing the possibility of nitrification and as we argue, this explanation is compatible with our observations.

*L 357 : If the authors envisage all possible CH4 sources in marine and freshwater environments then CH4 has more diverse sources that N2O. However, the only documented CH4 sources in tropical rivers are methanogenesis in riverbed and floodplain sediments.*

Even though we have removed marine sources from consideration in the revised manuscript (see earlier reviewer's comments and our response), that still leaves a number of potential methane production mechanisms in rivers: as we state in our original manuscript (new text lines 261-264), "Production by soil macrofauna (Kammann et al., 2009), archeal production related to plant productivity (Updegraff et al., 2001; Dorodnikov et al., 2011) and non-microbial, direct aerobic production, both by living plant tissue (Keppler et al., 2006; 2009) and in soils (Hurkuck et al., 2012, have all also been observed". While these various processes may not have been observed in tropical rivers ("*the only documented CH4 sources in tropical rivers are methanogenesis in riverbed and floodplain sediments,*- reviewer #1), we believe it scientifically unjustified to believe that they do not occur (absence of evidence is not evidence of absence!). Given the possibility of these other mechanisms we believe it remains true that methane potentially has a more diverse range of sources than does nitrous oxide in rivers.

*L 363-365 : The cited range of CH4 and N2O fluxes correspond to the basin average values and not the full range of individual CH4 and N2O flux estimates for each of the 12 river basins.*

*The range of CH4 fluxes for all individual estimates across the 12 rivers studied by Borges et al. (2015) is 0 to 274,600 µmol/m2/d for Aufdenkampe K estimate and 0 to 461,967 µmol/m2/d for Raymond K estimate. This range in fact corresponds to the one of the Congo that encompasses the data from all other African rivers. The range of N2O fluxes for all individual estimates across the 12 rivers studied by Borges et al. (2015) is -30 to 299 µmol/m2/d for Aufdenkampe K estimate and -37 to 377 µmol/m2/d for Raymond K estimate. This range in fact corresponds to the one of the Congo that encompasses the data from all other African rivers.*

We apologise for the error and we have substituted the correct ranges. The original text in lines 361- 365: "For 12 sub-Saharan African rivers Borges et al. (2015b) give ranges of 502 - 18019 µmol $CH_4$ $m^{-2}$ $d^{-1}$ and 2 - 16 µmol $N_2O$ $m^{-2}$ $d^{-1}$ using $k_w$ from Aufdenkampe et al. (2011), and 583 - 28579 µmol $CH_4$ $m^{-2}$ $d^{-1}$ and 2 - 28 µmol $N_2O$ $m^{-2}$ $d^{-1}$ using $k_w$ from Raymond et al. (2013)" now reads: "For 12 sub-Saharan African rivers Borges et al. (2015b) give ranges of 0 to 274,600 µmol $CH_4$ $m^{-2}$ $d^{-1}$ and -30 to 299 µmol $N_2O$ $m^{-2}$ $d^{-1}$ using $k_w$ from Aufdenkampe et al. (2011), and 0 to 461,967 µmol $CH_4$ $m^{-2}$ $d^{-1}$ and -37 to 377 µmol $N_2O$ $m^{-2}$ $d^{-1}$ using $k_w$ from Raymond et al. (2013)." (new text lines 374-377).

In view of this it has been necessary to modify the text in original lines 385-386: "The overall ranges of $CH_4$ and $N_2O$ emissions from rivers of the ROC (33-48705 µmol $CH_4$ $m^{-2}$ $d^{-1}$; 1-67 µmol $N_2O$ $m^{-2}$ $d^{-1}$) are somewhat wider than these earlier estimates for African and temperate rivers, the maximum values (Table 3) being around twice as high as previously reported".

This is now changed to: "The overall ranges of $CH_4$ and $N_2O$ emissions from rivers of the ROC (33 to 48705 µmol $CH_4$ m$^{-2}$ d$^{-1}$; 1 to 67 µmol $N_2O$ m$^{-2}$ d$^{-1}$, Table 3) fall within the ranges encompassed by these earlier estimates for African and temperate rivers" (new text lines 385-386). We also deleted "Nevertheless" from the start of the following sentence (new text line 387). We have also modified the corresponding line in the abstract, changing wider than previously estimated to "within the range previously estimated" (line 36).

**Anonymous Reviewer #2**

*..the dataset is valuable and merits publication – but I strongly agree with Reviewer #1 that it is crucial that the full data are made available in digital form, there is little point in collecting these data and leaving a legacy of just average or mean values in a summary Table.*

We agree entirely; please see our response to Reviewer #1.

*-Title: I find the title a little awkward, the 'riverine source of tropospheric CH4 and N2O' is somewhat misleading as the paper does not focus on budgeting sources of tropospheric CH4 and N2O, it looks at CH4 and N2O exchange (source, or sink insome cases for N2O) between surface waters and atmosphere. I suggest to rephrasethe title.*

While we do not agree at all that the title as it is implies some aspect of budgeting sources of tropospheric $CH_4$ and $N_2O$, we agree to remove the contentious word "tropospheric", so that the revised title now reads "The riverine source of $CH_4$ and $N_2O$ from the Republic of Congo, Western Congo Basin". We contend that this is now clear and unambiguous.

*-L19-20: 'predominantly supersaturated': with an average of 100% saturation, why call this 'predominantly oversaturated' ? -L22: The abbreviation ROC is used here for the first time – spell it out here.*

This is because they were predominantly supersaturated in savannah rivers: most of the samples were >100% saturation but that is not to say that the average saturation cannot be 100%. It is simple statistics. Consider a simple example of three river samples, of 70%, 116% and 115% saturation respectively. The average ± standard deviation of these is 100 ± 26% but based on the samples collected, these river waters as a group are predominantly supersaturated because two thirds of them have > 100% saturation.

We have added "(ROC)" immediately following "Republic of Congo" in new text line 15.

*-L32-33: awkward sentence, not clear what the authors are trying to say here ('was coincident with')*

We see absolutely nothing awkward with the sentence. It is grammatically sound and we see no reason to change it.

*-L70: what do you mean with 'seasonal emissions' ?*

Again, we believe this to be quite clear. It is a term that is in frequent use in the literature.

*-L124: inoculated: poisoned*

We believe this to be a rather trivial criticism. We have used both terms in previous papers in this context and both are acceptable.  However, we have replaced inoculated by poisoned (new text line 134)  to satisfy the reviewer's preference.

*-L240-264: I suggest deleting this entire section, not relevant for freshwater systems as pointed out by Reviewer #1.*

This has indeed been modified to remove material that is not relevant to freshwaters; please see response to Reviewer #1.  However, as our modified text shows, we do not agree that the entire deletion of this section is at all appropriate because some aspects of it are indeed relevant to freshwaters and it is therefore important to retain the modified text that we discuss above, in our response to reviewer #1, i.e. new text lines 259-264.

**L. Golston**

*L41-42: Hartmann et al. 2013 does not describe the impacts of CH4 on oxidizing capacity and other atmospheric species. Myhre et al. 2013 is a suitable reference.*

We have now substituted (Myhre et al., 2013) for (Hartmann et al., 2013) in new text line 42

*L47-48, also L53-54: This is the global surface mean, not the tropospheric mean, which would be slightly lower. Chapter 2 of AR5 (Hartmann et al. 2013) is a better citation than Chapter 6 (Ciais et al. 2013) since it details the data sources where these numbers are derived from. Also, simply say "150%" rather than "more than 150%".*

We have substituted (Hartmann et al., 2013) for (Ciais et al., 2013) as suggested (new text lines 53 and 58).  We have also deleted "more than" in original text line 48 (new text line 52).

*L48-49: It is not clear why "soils, freshwaters and coastal waters" is given. Freshwater and coastal waters were not explicitly discussed in the reference, and this excludes important sources they do give including animal waste management, biomass burning, oceans, and energy systems.*

On reflection we have decided that in order to avoid any unnecessary confusion it is best to remove the sentence in question, in lines 48-50: "Increasing tropospheric $N_2O$ largely reflects its enhanced emission from soils, freshwaters and coastal waters via the accelerated mobilisation of reactive nitrogen (Syakila and Kroeze, 2011)". Doing so does not materially affect our overall argument. We have also deleted the citation to (Syakila and Kroeze, 2011) from the reference list as it is not cited elsewhere in the manuscript.

*L50-51: This has several issues and does not follow from the previous sentence. 1) rivers, estuaries, and coastal zones was already included as a source category for N2O in IPCC AR4 (see Denman et al. 2007), though the sentence seems to imply it was added in AR5. 2) it is not that they switched the classification from natural to anthropogenic, but that sufficiently reliable data became available on the anthropogenic component.*

We have removed the previous sentence so the first comment no longer applies. We would like to retain this information about the switch in classification from natural to anthropogenic but we accept this point and so we have modified the sentence: "Consequently the IPCC now classifies river, estuary and coastal zone $N_2O$ sources as anthropogenic (Ciais et al., 2013)" (original text lines 50-51) and we have replaced it with: "Of particular note in regard to $N_2O$, the availability of sufficiently reliable data on the anthropogenic components of river, estuary and coastal zone sources resulted in a change in their classification in the IPCC AR4 synthesis, from "natural" to "anthropogenic" (Denman et al., 2007) (new text lines 53-55). We have also added (Denman et al., 2007) to the reference list.

*L60-61: "based on warming potentials and atmospheric lifetimes" -> "based on a 100year Global Warming Potential" would be more descriptive*

We have amended the text as suggested. "Converting it to $CO_2$ equivalents based on warming potentials and atmospheric lifetimes" (original text lines 61-62) now becomes "Converting these to $CO_2$ equivalents based on a 100 year Global Warming Potential" (new text line 70).

*L60-62: Kirschke et al. 2013 gives a bottom-up value of 40 [8-73] Tg/yr for freshwater sources, and a total source of 678 [542-852] Tg/yr, giving a mean freshwater contribution of about ~6% along with a wide uncertainty range. The estimates given of ~30-47% for natural and ~12-20% total sources seem to be miscalculated. In addition, it is unclear why we are asked to ignore the range of 10 – 100 Tg/yr given on L57 for freshwater, which would yield an even larger range of uncertainty.*

Considering the first statement that *Kirschke et al. (2013) give a bottom-up value of 40 [8-73] Tg/yr for freshwater sources*, this is 4 X $10^{13}$ g $yr^{-1}$ and the range (8-73 Tg/yr) is 0.8-7.3 x $10^{13}$ g $yr^{-1}$. Therefore this is already encapsulated in our statement that "The global freshwater $CH_4$ source could be ~$10^{13}$-$10^{14}$ g $yr^{-1}$ (Bastviken et al., 2011; Kirschke et al., 2013; Stanley et al., 2016)" (new text lines 61-62) so we see no issue here, although we have now modified this statement in light of information discussed below.

The assertion that: *a total source of 678 [542-852] Tg/yr, giving a mean freshwater contribution of about ~6%* is based entirely on the estimates presented in Kirschke et al., (2013), whereas we have additionally accounted for the estimates presented in Bastviken et al. (2011) and Stanley et al. (2016). However, we do accept that our calculated ranges based on these are a bit out; the lower ends are an order of magnitude too high, which must have been a transcription error, because the upper ends are about right given the errors. To do this calculation we took the maximum and minimum estimates for total and natural emissions presented in Kirschke et al. (2013) (Natural emissions 179-484 x $10^{12}$ g $yr^{-1}$; Total emissions 526-852 x $10^{12}$ g $yr^{-1}$). We then divided the minimum of our freshwater range ($10^{13}$ g $yr^{-1}$) by the maximum estimate in each case to give a lower boundary to the percentage contribution and we divided the maximum of our freshwater range ($10^{14}$ g $yr^{-1}$) by

the minimum estimate in each case to give an upper boundary to the percentage contribution.  This is quite usual practice that does not need spellig out in an Introduction. Our revised contributions are thus 2-56% for natural sources and 1-19% for total sources, i.e. the lower ends are an order of magnitude reduced but the upper ends only change marginally from the original estimates.

The comment: *In addition, it is unclear why we are asked to ignore the range of 10 – 100 Tg/yr given on L57 for freshwater, which would yield an even larger range of uncertainty* is misplaced because as we state above, these were taken account of in the calculation so are not ignored This comment seems to stem from a misreading of what we actually said.

Based on the above issues we have replaced the original text, incorporating the estimates of Kirschke et al (2103) for clarity. We have thus deleted" Notwithstanding the uncertainty, this freshwater source estimate is ~30-47 % of natural $CH_4$ emissions and ~12-20% of total $CH_4$ emissions (Kirschke et al., 2013)" (original text lines 60-61) and replaced it with" Notwithstanding this uncertainty, the global freshwater $CH_4$ source can be evaluated in the light of natural and total global $CH_4$ source estimates of 179-484 x $10^{12}$ g $yr^{-1}$ and 526-852 x $10^{12}$ g $yr^{-1}$ respectively (Kirschke et al., 2013). Quantifying the global freshwater contribution thus has high inherent uncertainty but based on these estimates it could be ~ 2-56 % of natural $CH_4$ emissions and ~1-19% of total $CH_4$ emissions" (new text lines  66-70).

*L63: Please double check this. It does not seem to be consistent with Figure 2 in Le Quéré et al. 2015 which suggests a combined sink of ~5.5 x 10ˆ15 g C yr-1*

This seems to be a typographic error that we have now corrected (new line 72) but the conclusion of a significant offset is unchanged.

*L159-160: Same comment about being surface rather than tropospheric mean CH4 and N2O. Also unclear how to access the referenced data using the link given at [http://www.eea.europa.eu/data-and-maps/](http://www.eea.europa.eu/data-and-maps/)*

We have again deleted "tropospheric" (new text line 169), although we feel this a rather minor point. We have also replaced the original web-link with a more accessible data gateway: [http://cdiac.esd.ornl.gov/tracegases.html](http://cdiac.esd.ornl.gov/tracegases.html)  (new text line 170).

*L269: supersaturations -> super-saturations (for consistency with usage in the rest of the text)*

This has been changed as requested (new text line 279).

*L383: Melack et al. 2004 and Bastkviken et al. 2010 both missing from References list*

"Bastkviken et al. 2010" (original text line 384) is a typographic error. It should be Bastviken et al., 2011. This is now corrected (new text line 394).  Melack et al., 2004 is now added to the reference list.

*L396-398: It is clear there are issues with emissions estimation that need to be addressed in the future, but it is not clear with the issue is with CH4 and N2O measurement and data calibration. It seems that standard, accurate, techniques are available for collection of samples and subsequently measuring CH4 and N2O concentrations please remove or clarify this point.*

We strongly disagree! To state that *standard, accurate, techniques are available for collection of samples and subsequently measuring CH₄ and N₂O concentrations* is both naïve and too simplistic. The issue of $N_2O$ and $CH_4$ calibration is extremely important and it is far from resolved. There are no currently agreed international calibration standards for these gases, individuals currently making or obtaining their own by various means. To rectify this two of us (RCUG and JB) are members of an international SCOR working group on $N_2O$ and $CH_4$, one of the aims of which is the full standardization of calibration gases and associated protocols. A recent inter-calibration exercise (results in prep.) demonstrates clearly the discrepancies that can in large part be attributed to errors in calibration. Having said that we have modified the text in question in order to emphasize our point and to publicize the SCOR effort, which encourages dialogue with interested parties. We have therefore deleted: "This is an issue that is yet to be adequately addressed, not only for freshwaters but for aquatic systems more generally". (original text lines 398-399) and replaced it with "There are currently no internationally agreed calibration standards for $CH_4$ or $N_2O$ but this is now being addressed via an international SCOR (Scientific Committee on Oceanic Research) Working Group (WG-143: https://portal.geomar.de/web/scor-wg-143/home), which is engaged in inter-laboratory calibration and the dissemination of high quality calibration gases. WG-143 welcomes additional interest from the wider aquatic $CH_4$ and $N_2O$ research community" (new text lines 408-413).

---

## Editor Decision (ED1)

**Abstract** *(with insertions/deletions):*

We report on concentrations of dissolved $CH_4$, $N_2O$, $O_2$, $NO_3^-$ and $NH_4^+$, and corresponding $CH_4$ and $N_2O$ emissions for river sites in savannah, swamp forest and tropical forest, along the Congo main stem and in several of its tributary systems of the Western Congo Basin, Republic of Congo (ROC), during November 2010 (41 samples; "wet season") and August 2011 (25 samples; "dry season"; $CH_4$ and $N_2O$ only). Dissolved inorganic nitrogen (DIN: $NH_4^+ + NO_3^-$; wet season) was dominated by $NO_3^-$ (63 ± 19% of DIN), total DIN concentrations (1.5-45.3 µmol $L^{-1}$)  are consistent with  near absence of agricultural, domestic and industrial sources. *Question: Is this true for all the three land types?* Dissolved $O_2$ (wet season) was mostly under-saturated in swamp forest (36 ± 29%) and tropical forest (77 ± 36%) rivers but predominantly super-saturated in savannah rivers (100 ± 17%). The dissolved concentrations of $CH_4$ and $N_2O$  are within  the range of values reported earlier for sub-Saharan African rivers.  Dissolved $CH_4$ was  found to be super-saturated (11.2 - 9553 nmol $L^{-1}$; 440-354400% *Comment: Check this number);* whereas $N_2O$ ranged from strong under-saturation to  super-saturation (3.2-20.6 nmol $L^{-1}$; 47-205%). Evidently, rivers of the ROC are persistent local sources of  $CH_4$  and can be  a minor source or sink for $N_2O$. During the dry season,  mean and range of $CH_4$ and $N_2O$ concentrations were  quite similar for all the three land types whereas seasonal differences in the mean and range were not significant for $N_2O$ concentration  in any land type or for $CH_4$ in savannah rivers. The latter observation is consistent with seasonal buffering of river discharge by an underlying sand-sandstone aquifer.  In contrast, $CH_4$ concentration in swamp and forest river  was significantly higher in the wet season,  suggesting that $CH_4$ can be derived from floating macrophytes during flooding and/or enhanced methanogenesis in adjacent  soils in flood bank. Swamp rivers also exhibit both low (47%) and high (205%) $N_2O$ saturation but wet season values were overall significantly lower than in either tropical forest or savannah rivers. These  rivers were always super-saturated (103-266%) and for which the overall mean and range of $N_2O$ were not significantly different. In swamp and forest rivers  $O_2$-saturation (%) -varied  inversely with  $CH_4$ saturation (log %) and  linearly with  $N_2O$ saturation (%).  A significant positive correlation for $N_2O$ - $O_2$  saturation in swamp rivers  is consistent with  $N_2O$ and $O_2$ under-saturation, indicating $N_2O$ consumption  during denitrification in the sediment.  In savannah rivers persistent $N_2O$ super-saturation and a negative $N_2O$ - $O_2$ correlation  suggest $N_2O$ production mainly by nitrification, consistent with  significant correlation between $N_2O$ and $NH_4^+$ than between $N_2O$ and $NO_3^-$. Our range  of values for $CH_4$ and $N_2O$ emission fluxes (33-48705 µmol $CH_4$ $m^{-2}$ $d^{-1}$; 1-67 µmol $N_2O$ $m^{-2}$ $d^{-1}$) are within the range previously estimated for sub-Saharan African rivers  and associated with uncertainties  arising from our use of "basin-wide" values for $CH_4$ and $N_2O$ gas transfer velocities.  Furthermore, as we did not account for any contribution from ebullition, which is quite likely for $CH_4$  (at least 20%), our emission fluxes  for $CH_4$ are  rather conservative estimates.

**Abstract** *(with all make up):*

We report on concentrations of dissolved $CH_4$, $N_2O$, $O_2$, $NO_3^-$ and $NH_4^+$, and corresponding $CH_4$ and $N_2O$ emissions for river sites in savannah, swamp forest and tropical forest, along the Congo main stem and in several of its tributary systems of the Western Congo Basin, Republic of Congo (ROC), during November 2010 (41 samples; "wet season") and August 2011 (25 samples; "dry season"; $CH_4$ and $N_2O$ only). Dissolved inorganic nitrogen (DIN: $NH_4^+ + NO_3^-$; wet season) was dominated by $NO_3^-$ (63 ± 19% of DIN), total DIN concentrations (1.5-45.3 µmol $L^{-1}$) are consistent with near absence of agricultural, domestic and industrial sources. *Question: Is this true for all the three land types?* Dissolved $O_2$ (wet season) was mostly under-saturated in swamp forest (36 ± 29%) and tropical forest (77 ± 36%) rivers but predominantly super-saturated in savannah rivers (100 ± 17%). The dissolved concentrations of $CH_4$ and $N_2O$ are within the range of values reported earlier for sub-Saharan African rivers. Dissolved $CH_4$ was found to be super-saturated (11.2 - 9553 nmol $L^{-1}$; 440-354400% *Comment: Check this number)*; whereas $N_2O$ ranged from strong under-saturation to super-saturation (3.2-20.6 nmol $L^{-1}$; 47-205%). Evidently, rivers of the ROC are persistent local sources of $CH_4$ and can be a minor source or sink for $N_2O$. During the dry season, mean and range of $CH_4$ and $N_2O$ concentrations were quite similar for all the three land types; whereas seasonal differences in the mean and range were not significant for $N_2O$ concentration in any land type or for $CH_4$ in savannah rivers. The latter observation is consistent with seasonal buffering of river discharge by an underlying sand-sandstone aquifer. In contrast, $CH_4$ concentration in swamp and forest rivers was significantly higher in the wet season, suggesting that $CH_4$ can be derived from floating macrophytes during flooding and/or enhanced methanogenesis in adjacent flooded soils. Swamp rivers also exhibit both low (47%) and high (205%) $N_2O$ saturation but wet season values were overall significantly lower than in either tropical forest or savannah rivers. These rivers were always super-saturated (103-266%) and for which the overall mean and range of $N_2O$ were not significantly different. In swamp and forest rivers $O_2$-saturation (%) varied inversely with $CH_4$ saturation (log %) and linearly with $N_2O$ saturation (%). A significant positive correlation for $N_2O$ - $O_2$ saturation in swamp rivers is consistent with $N_2O$ and $O_2$ under-saturation, indicating $N_2O$ consumption during denitrification in the sediments. In savannah rivers persistent $N_2O$ super-saturation and a negative $N_2O$ - $O_2$ correlation suggest $N_2O$ production mainly by nitrification, consistent with significant correlation between $N_2O$ and $NH_4^+$ than between $N_2O$ and $NO_3^-$. Our range of values for $CH_4$ and $N_2O$ emission fluxes (33-48705 µmol $CH_4$ $m^{-2}$ $d^{-1}$; 1-67 µmol $N_2O$ $m^{-2}$ $d^{-1}$) are within the range previously estimated for sub-Saharan African rivers and associated with uncertainties arising from our use of "basin-wide" values for $CH_4$ and $N_2O$ gas transfer velocities. Furthermore, as we did not account for any contribution from ebullition, which is quite likely for $CH_4$ (at least 20%), our emission fluxes for $CH_4$ are rather conservative estimates.

---

## Author Response (AR2)

Response to comments of 09.02.17 by the Associate editor.

I have now made most of the additional changes that were suggested by the Associate Editor subsequent to his previous comments. All of these current changes relate to the abstract and are highlighted yellow in the revised document. While many are helpful and improved the text, I found that some others were less helpful and did rather less to enhance the text (although these have been incorporated). Some other suggested changes I chose not to adopt because either they altered the meaning of the text (which I want to avoid) or they introduced significant grammatical inconsistencies.

I was a little disappointed that these suggested changes are additional to issues highlighted previously and that they were not brought up as part of the last iteration. I would like to see this manuscript now move forward to publication so I trust that further iterations will not be required.

---

## Author Response (AR3)

Below are our final responses to the technical comments of the two reviewers. Reviewer comments are in italics, with our responses in normal font.

**Reviewer Report #1:**

*I had gone through the author replies earlier on, and am generally happy with the way the authors addressed comments and suggestions from the different reviewers.*
*A few technical suggestions for the supplementary material, however:*

*-add coordinates for the different sampling locations*

These have now been added to Supplementary Table S1.

*-use an appropriate number of decimals - e.g. there is little point in expressing %O2 with two decimals; and for N2O concentrations use a consistent number of deciments (now: data from some sites presented with one decimal, for other sites with two decimals).*

This has now been corrected in Supplementary Table S1 so that % $O_2$ no longer has decimals. All other data are now also consistently reported.

**Reviewer Report #2**

*Upstill-Goddard et al. have adequately addressed my concerns from my previous review. There a few minor issues listed below:*

*L65: Here and throughout please use gC or gCH4*

This has now been done

*L89: specify the number of tributaries*

We have now changed all references to "several" tributaries to "seven", this being the number of tributary systems listed in Table 1.

*L129: specify how was the water collected: Niskin bottle ?*

Yes these were collected using a Niskin-type sampler. We have now modified the text to include an additional sentence (lines 129-130), which reads: "Surface water samples (~0.3 m depth) were collected from central river channels using a standard "niskin"-type water sampler (http://www.tresanton.co.uk/standard.html)".

*L147: YSI is not a UK brand*

We have now modified this to "https://www.ysi.com/", which is the correct link.

*L355: Regarding N2O and DIN relations, Borges et al. (2015) envisaged this in several African rivers, refer to supplemental figures 5 and 6.*

We have modified this so that following "(Bouillon et al., 2012)", we now say "and in several other African rivers (Borges et al., 2015b)".

---

## Author Response (AR4)

This is the final version of this manuscript, which is now accepted for publication.  All suggested revisions have now been made to the satisfaction of the Associate Editor.

R.C. Upstill-Goddard  11 April 2017